# Leveraging Pre-trained Large Language Models to Construct and Utilize World Models for Model-based Task Planning

**Lin Guan** *
School of Computing & AI
Arizona State University
Tempe, AZ 85281
lguan9@asu.edu

**Karthik Valmeekam** *
School of Computing & AI
Arizona State University
Tempe, AZ 85281
kvalmeek@asu.edu

**Sarath Sreedharan**
Department of Computer Science
Colorado State University
Fort Collins, CO 80523
sarath.sreedharan@colostate.edu

**Subbarao Kambhampati**
School of Computing & AI
Arizona State University
Tempe, AZ 85281
rao@asu.edu

## Abstract

There is a growing interest in applying pre-trained large language models (LLMs) to planning problems. However, methods that use LLMs directly as planners are currently impractical due to several factors, including limited correctness of plans, strong reliance on feedback from interactions with simulators or even the actual environment, and the inefficiency in utilizing human feedback. In this work, we introduce a novel alternative paradigm that constructs an explicit world (domain) model in planning domain definition language (PDDL) and then uses it to plan with sound domain-independent planners. To address the fact that LLMs may not generate a fully functional PDDL model initially, we employ LLMs as an interface between PDDL and sources of corrective feedback, such as PDDL validators and humans. For users who lack a background in PDDL, we show that LLMs can translate PDDL into natural language and effectively encode corrective feedback back to the underlying domain model. Our framework not only enjoys the correctness guarantee offered by the external planners but also reduces human involvement by allowing users to correct domain models at the beginning, rather than inspecting and correcting (through interactive prompting) every generated plan as in previous work. On two IPC domains and a Household domain that is more complicated than commonly used benchmarks such as ALFWorld, we demonstrate that GPT-4 can be leveraged to produce high-quality PDDL models for over 40 actions, and the corrected PDDL models are then used to successfully solve 48 challenging planning tasks. Resources, including the source code, are released at: https://guansuns.github.io/pages/llm-dm.

## 1 Introduction

The field of artificial intelligence has been revolutionized with the advent of large pre-trained models. Of particular significance are transformer-based large language models (LLMs) which have showcased remarkable performance in natural language processing tasks. Along with these tasks, LLMs have been tested to perform another widely-studied crucial aspect of AI agents, namely, sequential decision-

---

*Equal contribution

37th Conference on Neural Information Processing Systems (NeurIPS 2023).

making or planning. Preliminary studies suggest that, in some everyday domains, LLMs are capable of suggesting sensible action plans [19, 1]. However, the correctness and executability of these plans are often limited. For instance, LLMs may regularly overlook the physical plausibility of actions in certain states and may not effectively handle long-term dependencies across multiple actions. Several approaches have been proposed to improve the planning capabilities of LLMs. One promising approach involves collecting feedback from the environment during plan execution and subsequently refining the plans. By incorporating various forms of feedback, such as sensory information [20], human corrections [60], or information of unmet preconditions [42, 56], the planners can re-plan and produce plans that are closer to a satisficing plan.

Despite the improvements in planning performance, LLMs are still far from being a usable and reliable planner due to various factors:

(a) LLMs have not yet demonstrated sufficient capabilities in reasoning and planning [24, 55, 53, 54, 31]. Recent investigations show that even when provided with detailed descriptions of actions, such as a PDDL domain model [33] or a natural-language version of a PDDL model, LLMs still struggle to produce correct and executable plans [48, 55].

(b) Existing LLMs-planning paradigms only allow for feedback collection in a fully online manner, meaning that the feedback signals are only available after the agent has started executing the plan. However, when a faithful simulator is not available or is expensive to use, collecting feedback through actual plan execution can be costly and may not fully exploit the advantages of provably sound planning, as seen in classical-planning literature [11, 13].

(c) LLMs exhibit complex behaviors that are not yet fully understood, particularly with respect to error occurrences. LLM planners are prone to repeating the same mistakes in slightly different scenarios. Repeatedly providing the same feedback can lead to frustration for end users.

To overcome these limitations, rather than using LLMs directly as planners, we advocate a model-based paradigm, wherein a PDDL world model is teased out of LLMs. We follow the identical problem setup as existing approaches, which involves providing the planner with a set of actions and their brief natural language descriptions. However, instead of directly mapping user commands to plans, we utilize LLMs to extract a symbolic representation of the actions in the form of PDDL action models. This intermediate output can be used with an external domain-independent planner to reliably search for feasible plans, or it can be used to validate and correct "heuristic" plans generated by an LLM planner. Additionally, our modular method essentially divides the planning process into two distinct parts, namely modeling the causal dependencies of actions and determining the appropriate sequence of actions to accomplish the goals. LLMs, which have been trained on extensive web-scale knowledge, exhibit greater proficiency in the former task rather than the latter.

Nevertheless, we still take into account the fact that the LLMs may not be able to generate error-free PDDL models at the outset. To address this, we show that LLMs can also serve as an interface between PDDL and any feedback sources that can provide corrective feedback in natural language, such as humans and the PDDL validator in VAL [18]. The LLM middle layer translates PDDL representation to natural language and presents it to users for inspection. The acquired feedback is then incorporated and archived back to the PDDL models. This conceals the complexity of PDDL from users who do not have prior knowledge of PDDL, and enables seamless inclusion of feedback. We conducted an extensive evaluation of our methodology on two IPC domains [22] from classical planning literature and a household domain that has a more diverse set of actions and constraints than commonly used benchmarks such as ALFWORLD [47]. We assess the quality of the generated PDDL models through manual evaluation. Results show that GPT-4 [37] generates high-quality PDDL domain models with over 400 literals for 41 actions in total. Then, by replaying and continuing the PDDL-construction dialogue, we show that GPT-4 can readily correct all the errors according to natural language feedback from PDDL validators and humans.

We consider two use cases of the generated PDDL action models for downstream planning tasks. For one, by utilizing an LLM to translate user instructions into goal specifications in PDDL [58, 30], we can use any standard domain-independent planner to search for a plan. On the other hand, the extracted PDDL model can be used to validate plans suggested by an LLM planner and to provide corrective feedback in the form of unmet preconditions or goal conditions. In this case, the PDDL model is essentially serving as an inexpensive high-level simulator or a human proxy to ensure plan correctness.

This reduces the reliance on faithful simulators or extensive manual inspection of plans by domain experts. Compared to the first approach, the second approach potentially offers better flexibility in incorporating both explicit and implicit user constraints in common-sense domains because of the LLM planner. For instance, the LLM planner can directly incorporate ordering constraints such as "heat the potato first before mashing it" and "bring me a fork first, then a plate." On the contrary, an approach purely based on classical planners would require extra steps, such as introducing extra state variables in the PDDL models, in order to accommodate such constraints. However, as demonstrated in our experiments, although the validation feedback significantly improves the plan correctness on average, the performance of the second approach is still limited by the "planning capability" of LLMs.

## 2 Related Work

**LLMs and planning.** The growing interest in evaluating the emergent abilities of LLMs paved way into exploring their abilities in sequential decision-making tasks. Preliminary studies [24, 55] have shown that off-the-shelf LLMs are currently incapable of producing accurate plans. But their plans can be used as heuristics or seeds to either an external planner or a human in the loop [55, 48]. SayCan [1] and Text2Motion [29] employ an LLM as a heuristic by utilizing it to score high-level actions, followed by a low-level planner that grounds these actions to determine the executability in the physical world. In a similar vein, [28, 50] use LLMs to generate plans represented in Python-style code. Other works have aimed to improve the planning performance of LLMs through prompt engineering [60] or collecting various forms of feedback such as sensory information [51, 20, 34], human corrections [60], self-corrections [46] or information of unmet preconditions [42, 56].

**Training transformers for sequential decision-making tasks.** Along with using off-the-shelf LLMs, there are works that either fine-tune LLMs [55, 38] or train sequence models [62, 27, 7, 43] for sequential decision making tasks. Experiments in [26] have shown that training sequence models on a specific task gives rise to an internal world representation within the model. In this work, we use off-the-shelf LLMs to construct symbolic world models without performing any extra training.

**Learning/acquiring symbolic domain models.** In classical planning, the community has explored numerous learning-based methods [59, 61, 9, 25, 4] and interactive editor-based methods [49] for acquiring symbolic domain models. For a more comprehensive survey, we refer the reader to [2, 6]. Here, we are interested in leveraging the common-world knowledge embedded in LLMs and their in-context learning ability for constructing domain models. Recent studies have shown the efficacy of LLMs in translating natural language to formal descriptions [35] or constructing PDDL goals from natural-language instructions [58, 32]. Moreover, a contemporary work [15] considers the use of LLM as a parametric world model and plan critic. However, unlike a symbolic model that can simulate plan outcomes with guaranteed correctness, using LLMs directly as a world model actually adds another layer of errors. There is evidence that autoregressive models lack reliable capacity for reasoning about action effects [3, 31] and capturing errors in candidate plans [53, 54].

**Language models with access to external tools.** Since LLMs are approximately omniscient, they may not always outperform specialized models or tools in specific downstream tasks. To address this limitation, frameworks have been developed to enable LLMs to utilize external tools for performing sub-tasks like arithmetic [45] and logical reasoning [39, 57]. In this context, our work can be regarded as an exercise in employing external sound planners to augment the capacity of LLMs for more reliable plan generation.

## 3 Problem Setting and Background

Our work focuses on a scenario where an intelligent agent receives high-level instructions or tasks, denoted as $i$, from a user. The agent is capable of only executing skills or operations that are part of a skill library $\Pi$, where each skill $k$ has a short language description $l_k$. We assume that the agent is equipped with the low-level control policies corresponding to these high-level skills. In order to achieve the goal conditions specified in $i$, a planner, which can be either an LLM or an external planner [16, 12, 17], needs to come up with a sequence of high-level skills that the agent can execute. This type of problem is referred to as a sequential decision-making or planning problem. Similar to previous works such as [60, 20], we also allow for human-in-the-loop feedback during both the domain-model construction and plan execution stages. In the next subsections, we describe the formalism behind planning problems and a standard way in the literature to specify them.

## 3.1 Classical planning problems

The most fundamental planning formalism is goal-directed deterministic planning problem, referred to as a classical planning problem in the planning literature. A classical planning problem [44] can be formally represented with a tuple $\mathcal{P} = \langle \mathcal{D}, \mathcal{I}, \mathcal{G} \rangle$. $\mathcal{D}$ is referred to as the domain, $I$ is the initial state, and $\mathcal{G}$ is the goal specification. The state space of a planning problem consists of the truth assignments for predicates. The domain $\mathcal{D}$ is further defined by the tuple $\mathcal{D} = \langle \mathcal{F}, \mathcal{A} \rangle$. $\mathcal{F}$ corresponds to the set of fluents, i.e., the state variables used to define the state space with each fluent corresponding to a predicate with some arity. $\mathcal{A}$ corresponds to the set of actions that can be performed. Each action $a_i[\mathcal{V}] \in \mathcal{A}$ (where $\mathcal{V}$ is the set of variables used by the operator $a_i$ and each variable could be mapped to an object) can be further defined by two components, the precondition $\texttt{prec}[\mathcal{V}]$ which describes *when* an action can be executed, and the effects $\texttt{eff}[\mathcal{V}]$ which defines *what happens* when an action is executed. We assume that $\texttt{prec}[\mathcal{V}]$ consists of a set of predicates defined over the variables $\mathcal{V}$. An action is assumed to be executable only if its preconditions are met, i.e, the predicates in the precondition hold in the given state. The effect set $\texttt{eff}[\mathcal{V}]$ is further defined by the tuple $\langle \texttt{add}[\mathcal{V}], \texttt{del}[\mathcal{V}] \rangle$, where $\texttt{add}[\mathcal{V}]$ is the set of predicates that will be set true by the action and $\texttt{del}[\mathcal{V}]$ is the set of predicates that will be set false by the action. An action is said to be grounded if we replace each of the variables with an object, else it is referred to as a lifted action model. A solution to a planning problem is called a plan, and it is a sequence of actions that once executed in the initial state would lead to a state where the goal specification holds. Classical planning problems are one of the simpler classes in planning and there are multiple extensions with more complex forms of preconditions, conditional effects, and also support for richer planning formalisms.

## 3.2 PDDL

Planning Definition and Domain Language (PDDL) [33], is the standard encoding language for classical planning problems. Here is an example of a lifted action in PDDL which corresponds to putting a block onto the table in the classical Blocksworld domain:

```
(:action PutDownBlock
    :parameters (?x - block)
    :precondition (and (robot-holding ?x))
    :effect (and (not (robot-holding ?x)) (block-clear ?x) (robot-hand-empty) (block-on-table ?x)))
```

The parameters line provides the possible variable(s), and in this case, ?x represents the block to put down. The precondition states that the robot must be holding the block in its gripper. The effects line describes the expected outcome of this action.

# 4 Methodology

PDDL provides a succinct and standardized way to represent a world model. Once a PDDL model is constructed, it can be seamlessly used by any domain-independent planner developed in the automated planning community to search for a plan given the initial state and goal conditions. In this section, we will introduce our solution for constructing PDDL models using LLMs. We then discuss techniques for correcting errors in the generated PDDL models. Finally, we present the full pipeline for utilizing the generated PDDL models to solve planning problems.

## 4.1 Constructing PDDL models with LLMs

Our approach involves prompting pre-trained LLMs with the following information: (a) detailed instructions for the PDDL generation task, outlining components of upcoming inputs and desired outputs; (b) one or two examples from other domains (e.g., the classical Blocksworld domain) for illustrating the input and output formats; (c) a description of the current domain, including contextual information about the agent's tasks and physical constraints due to the specific embodiment of the agent; (d) a description of the agent's action; and (e) a dynamically updated list of predicates that the LLM can reuse to maintain consistent use of symbols across multiple actions. Note that *the predicate list is initialized to an empty list*, and thus all predicates are introduced by the LLM. The structure of the prompt is illustrated in Fig. 2, and a complete prompt for the household-robot domain can be found at Appx. A.6.1.

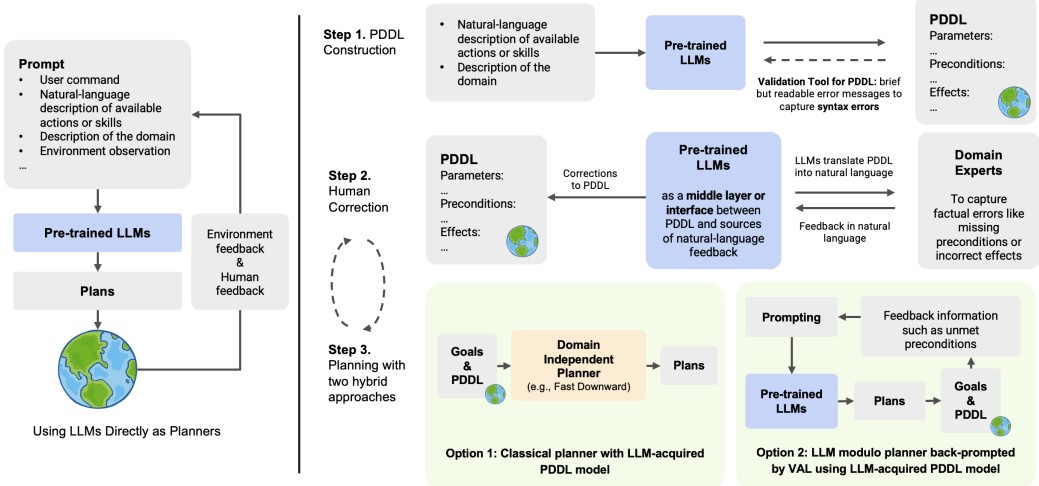

Figure 1: An overview of our framework and existing methods that use LLMs directly as planners.

Depending on the information included in the action description or the domain context, users may gain varying levels of control over the extracted PDDL or receive differing levels of support from the LLMs. On one hand, when the user provides only a minimal description of the action, such as "this action enables the robot to use a microwave to heat food," we not only use the LLM as a PDDL constructor but also leverage the common world knowledge encoded within the model for knowledge acquisition. This is particularly useful when expanding the set of actions for an AI agent. For example, a robot engineer could set up a training environment for skill learning by following the suggested preconditions and effects. On the other hand, when some preconditions or effects are explicitly mentioned in the prompt, we rely more on the LLM's ability to parse the knowledge provided in natural language and to precisely represent it by devising a collection of predicates. This capability is useful when there could be different initial setups of a skill, and the engineers have already made some assumptions on the preconditions at the time of designing the skill. This capability is also crucial when constructing PDDL for specialized domains. For instance, robots such as Fetch and Spot Robot have only one robot arm, which is less flexible than a human arm, and are therefore subject to many uncommon physical constraints.

The desired output comprises the following elements: (a) the list of arguments for the action; (b) the preconditions and effects expressed in PDDL; and (c) a list of any newly defined predicates and their descriptions in natural language, if applicable. An example output is shown in Fig. 2. Our algorithm generates PDDL models for each action separately, one at a time, by iterating over the set of actions. Any newly defined predicates will be added to an actively maintained predicate list, such that the LLM can reuse existing predicates in subsequent actions without creating redundant ones. Once we obtain the initial PDDL models and the full predicate list, we repeat the entire process but with all of the extracted predicates presented to the LLM. Running the generation process twice is useful because the LLMs may be unaware of some precondition(s) during the first iteration, especially if the precondition(s) are not explicitly mentioned. For instance, the LLM may overlook the fact that a furniture piece can be openable, but a predicate created in the "open a furniture piece or appliance" skill can inform the LLM of this fact. One alternative to this action-by-action generation could be to include descriptions of all the actions in the prompt and require the LLM to construct the entire domain model in a single dialogue. An additional discussion on this can be found at Sec. A.2 in Appendix.

It is worth noting that every time a new predicate is defined, the LLM is required to give the natural language description of it. As we will see in the following sections, this is crucial for enabling any user to easily understand and inspect the generated PDDL models without having to delve into the low-level symbolic representation. Additionally, natural language descriptions allow the predicate values of the initial state to be automatically grounded by using LLMs to translate environment description in natural language to PDDL [30], or leveraging pre-trained vision-language models

```
Instructions for the PDDL generation task
You are defining the preconditions and effects (represented in PDDL format) of an AI agent's
↪   actions. Information about the AI agent will be provided in the domain description ...
One or two examples from other domains for illustrating the input and output formats
Here are two examples from the classical BlocksWorld domain for demonstrating the output format.
...
Here is the task.
A natural language description of the domain
Domain information: The AI agent here is a household robot that can navigate to various large and
↪   normally immovable furniture pieces or appliances in the house to carry out household tasks
↪   ...
A natural language description of the action
Action: This action enables the robot to toggle small appliances (like humidifiers and light
↪   bulbs) which are toggleable to switch them on ...
The dynamically updated list of predicates
You can create and define new predicates, but you may also reuse the following predicates:
1. (robot-at ?r - robot ?f - furnitureAppliance): true if the robot ?r is at the furniture or
↪   appliance ?f
2. (object-in-on ?o - householdObject ?f - furnitureAppliance): true if the object ?o is in or on
↪   the furniture or appliance ?f
...
Parameters:
-------------------------------------
The LLM:
...
2. ?o - householdObject: the small appliance to be toggled on
...

Preconditions:
(and
    ...
    (not (appliance-on ?o))
)

Effects:
(and
    (appliance-on ?o)
)

New Predicates:
1. (appliance-on ?o - householdObject): true if the small appliance ?o is switched on
```

Figure 2: The prompt template for PDDL construction and an example of the LLM output for the household domain.

[41, 37, 10] and querying them in a question-answering manner, based on observations from the environment.

## 4.2 Correcting errors in the initial PDDL models

As with any use case involving LLMs, there is no guarantee that the output is completely error-free. Therefore, it is essential to incorporate error-correction mechanisms. While it may be easy for PDDL experts to directly inspect and correct the generated PDDL models, we cannot assume that all end users possess this level of expertise. Our solution is to use the LLM *as a middle layer or interface* between the underlying PDDL model and any feedback source that can provide corrective feedback in natural language. We consider two feedback sources in this work, namely the PDDL model validation tools (e.g., the one in VAL [18]) and human domain experts. The former is used to detect basic syntax errors, while the latter is mainly responsible for catching factual errors, such as missing effects. It is worth noting that the feedback sources are not limited to those mentioned above, and we leave the investigation of other sources for future research.

For corrective feedback from PDDL validators, a generated PDDL model is directly presented to the validator to obtain brief but readable error messages. Examples of feedback messages for syntax errors are shown in Appx. A.3. For corrective feedback from users, a PDDL model is translated into its natural-language version based on the natural language descriptions of the predicates and parameters (Sec. 4.1). The user can then examine potentially erroneous action models. Human corrections can occur both during the construction of PDDL models and after the models have been used for planning. Although there are techniques available to assist users to locate errors in the models (as discussed in

Appx. A.4), this is beyond the scope of this work, since the focus here is to investigate the feasibility of using LLMs to correct PDDL models based on feedback. We also note that correcting action models is not more cognitively demanding than correcting plans or the "reasoning traces" of an LLM planner [60]. In fact, when correcting plans, humans must also maintain the action models and their causal chains in mind in order to validate the plans. More importantly, once the action models are corrected, users no longer need to provide similar feedback repeatedly. Finally, corrective feedback is integrated by replaying and continuing the PDDL-construction dialogue. Examples of such dialogues can be found in Sec. A.7, Sec. A.9, and Sec. A.11 in Appendix.

### 4.3 Generating plans with the extracted PDDL models

Recall that given the set of extracted predicates and their natural language descriptions, we can get the grounded initial state by using LLMs to translate descriptions of the environment to PDDL, or by observing the environment and querying pre-trained vision-language models. Besides, the goal specification can be obtained by using an LLM to parse the user's command and convert it into a symbolic form, as done previously in [30, 58, 32]. With this setup, the following two methods can be used to generate the final plans.

**Classical planner with LLM-acquired PDDL model.** One straightforward approach is to employ a standard domain-independent planner to reliably find a satisficing or even optimal plan for the specified goal. In common-sense domains where LLMs may generate meaningful "heuristics", the LLM plans may also be used as seed plans for a local-search planner such as LPG [12] to accelerate the plan searching. This is similar to the approach suggested in [55], but with a higher degree of automation.

**LLM modulo planner backprompted by VAL using LLM-acquired PDDL model.** As outlined in Sec. 1, we can also use the extracted PDDL as a symbolic simulator or human proxy to provide corrective feedback based on validation information to an LLM planner. With this setup, the planner can iteratively refine the plans through re-prompting [42].

It is worth noting that depending on the specific problem settings, the extracted PDDL model can also be used for tasks other than task planning. For instance, in cases where reinforcement learning is permissible, the domain model can be used to guide skill learning [21, 8] or exploration even if the model is not fully situated [14].

## 5 Empirical Evaluation

We conduct our experiments [2] on an everyday household-robot domain and two more specialized IPC domains (i.e., Tyreworld and Logistics). The Household domain is similar to other commonly used benchmarks like ALFWORLD [47] and VirtualHome [40]. However, in our household domain, a single-arm robot is equipped with a more diverse and extended set of 22 mobile and manipulation skills. In addition, we apply more rigorous physical-plausibility constraints to each skill. A detailed description of this domain can be found at Appx. A.5. In our experiments, we first evaluate the quality of PDDL models generated by the LLMs. Next, we assess the ability of the LLMs to incorporate corrective feedback from both PDDL validators and users in order to obtain error-free PDDL models. Lastly, we showcase multiple ways to use the corrected PDDL model for downstream planning tasks. We present the results of GPT-4 [37] and GPT-3.5-Turbo [36] for PDDL construction (we also conducted experiments with GPT-3 [5], and observe that its performance is comparable to that of GPT-3.5-Turbo).

### 5.1 Constructing PDDL

In PDDL construction tasks, we aim to investigate the extent to which LLMs can construct accurate PDDL models before getting corrective feedback from domain experts. For all the domains, two actions from the classical Blocksworld domain are used as demonstrations in the prompt so that the end user is not required to come up with any domain-specific example. To evaluate the degree of correctness, we recruit multiple graduate students who possess expertise in PDDL. These experts are responsible for annotating and correcting any errors present in the generated PDDL models. As an evaluation metric, we count and report the total number of annotations, which may include the removal of irrelevant preconditions, the addition of missing preconditions, the replacement of incorrect predicates, the inclusion of missing parameters, and other commonly made corrections. Note

---

[2]Experiments were done in May 2023.

that the number of annotations can be viewed as the approximate distance between a generated PDDL model and its corrected version. In order to provide the reader with a comprehensive understanding of the quality of the generated models, we also list all the models and collected annotations in Appendix. In each of the figures, errors that affect the functionality of the PDDL model are highlighted in yellow, while minor issues are highlighted in green. One example of a minor issue is the redundant inclusion of (`pickupable ?o`) in preconditions when (`robot-holding ?o`) has already been listed. The former is unnecessary because it can be implied by the latter, but this only affects conciseness rather than functionality.

| Domain | # of actions | # of params and literals | # of GPT-4 errors | # of GPT-3.5-Turbo errors |
|---|---|---|---|---|
| Household | 22 | 271 | 53 | 218+ |
| Logistics | 6 | 54 | 2 | 38 |
| Tyreworld | 13 | 108 | 4 | 94+ |

Table 1: The number of errors in the domain models produced by the LLMs for each of the domains. A "+" mark indicates that the generated model is excessively noisy, making it challenging to determine an exact number of errors.

We first evaluate the PDDL models generated when partial constraint information is given, as this is closer to most of the practical use cases where constraints on skills in the library $\Pi$ are often pre-specified. In this setting, our evaluation focuses on the LLMs' ability to accurately recover a "ground truth PDDL" that captures the mentioned constraints and underlying dependencies among skills. Our results indicate that GPT-4 can produce high-quality PDDL models with significantly fewer errors when compared to GPT-3.5-Turbo. Table 1 presents the number of errors in the generated domain models for each domain. To help the readers understand the complexities of the action models, we additionally report the total number of parameters and literals in the final corrected domain models produced by GPT-4. Out of the total 59 errors made by GPT-4, three of them are syntax errors and the rest are factual errors such as missing preconditions and effects. This observation suggests that while GPT-4 demonstrates proficiency in adhering to the grammar of PDDL, it may still have an inaccurate understanding of the actions. By examining the set of predicates (listed in the Appendix), we also find that GPT-4 can devise a set of intuitively-named predicates that can concisely and precisely describe the states of objects and events in the domain. In contrast, GPT-3.5-Turbo produces highly noisy outputs with over 350 errors. This suggests that our framework relies heavily on GPT-4's improved capability in understanding symbols, and future work may investigate how to enable the use of more lightweight models (e.g., by fine-tuning on some PDDL datasets). Furthermore, recall that when the action description contains minimal information, LLMs could also be utilized to propose preconditions and effects to assist with knowledge acquisition. To verify this hypothesis, we conduct additional experiments on the Household domain that can have a more open-ended action design. In this setting, the correctness of the action models is determined based on whether the preconditions and effects establish correct connections among the actions. Our results show that GPT-4 can suggest meaningful action models, and the generated PDDL models have only around 45 errors.

Although GPT-4 has shown improved performance in the PDDL construction task, our experiments still uncover some limitations. Firstly, GPT-4 still exhibits a shallow understanding of the causal relationships between actions, particularly when it comes to tasks involving reasoning skills such as spatial reasoning. For instance, when constructing the model of action "pick up an object from a furniture piece," GPT-4 fails to consider that there could be other objects stacked on top of the target object, even if relevant predicates are provided (which were created in the action "stack objects"). In addition, although it occurs rarely, GPT-4 may output contradictory effects. For instance, in the action of mashing food with a blender, GPT-4 lists both (`not (object-in-receptacle ...)`) and (`object-in-receptacle ...`) as effects at the same time.

## 5.2 Correcting PDDL with domain experts

We proceed with the PDDL models generated by GPT-4 when the constraint information is partially given. Our objective is to demonstrate the feasibility of using GPT-4 as a middle layer to incorporate natural-language feedback and correct the PDDL models. As discussed in Sec. 4.2, we use PDDL validators to capture basic syntax errors. In the Household domain, there are two syntax errors

associated with improper usage of relevant predicates due to issues with the object types of parameters [3]. As shown in Appx. A.7.1, by continuing the PDDL-construction dialogue with a feedback message "`the second parameter of object-on should be a furnitureAppliance but a householdObject was given,`" GPT-4 can locate the inaccurate PDDL snippet and replace it with a correct one. For the other factual errors, GPT-4 successfully corrects all of them based on the natural language feedback. An example feedback message on factual errors is "`there is a missing effect: the item is no longer pickupable after being mashed.`" More PDDL-correction conversations can be found in Appendix. We also experiment with feedback written in various ways, and GPT-4 is able to understand all the messages and successfully correct the models. To quantify how effectively GPT-4 utilizes feedback from domain experts, we count the number of feedback messages concerning factual errors. Our result shows that GPT-4 required 59 feedback messages to address a total of 56 factual errors. There are three instances where additional feedback was needed. One case involved the user reiterating the error, while the other two cases involved GPT-4 introducing new errors. Furthermore, we attempt to correct the same errors using GPT-3.5-Turbo. Results show that GPT-3.5-Turbo not only fails to correct all the errors but also occasionally introduces new errors, again confirming its lack of ability to manipulate symbols. Some examples can be found in Appendix starting from Sec. A.7.3.

## 5.3 Generating plans with the extracted PDDL models

For planning tasks (i.e., user instructions and initial states), we use the Household domain and Logistics domain, where state-of-the-art LLM planners struggle to find valid plans. We sampled 27 tasks for Household and 21 for Logistics. For the initial states, we assume the grounding is provided, and for the goals, we leverage GPT-4 to translate user instructions into PDDL goal specifications in terms of the extracted predicates (an example prompt can be found at Appx. A.13), and send it over to a standard STRIPS planner *which already has access to the domain model acquired through LLMs*. With this setup, a classical planner Fast Downward [16] can effectively find valid plans in 95% of the cases (the failures were only due to goal translation errors). Note that in contrast to earlier methods such as [30] that use LLMs only as a mechanism for translating user goals to PDDL format, and throw that over to external sound planners with hand-crafted correct PDDL domain models, our approach uses LLMs themselves to develop the PDDL world model driving the external planner.

On the other hand, for the approach that utilizes PDDL models to validate LLM plans (i.e., LLM modulo planner back-prompted by VAL using LLM-acquired domain model), we employ the state-of-the-art algorithm ReAct [60] with GPT-4 as the underlying LLM planner. However, we made two modifications to the prompt design. Firstly, we provide a detailed description of all actions in natural language, including parameters, preconditions, and effects. These descriptions are ob-

| Planner Type | Household | Logistics |
|---|---|---|
| Only LLM Planner | 15% | 0% |
| Fast Downward with LLM-acquired PDDL model | 95% | 100% |
| LLM backprompted by VAL using LLM-acquired PDDL model | 48% | 33% |

Table 2: Success rates of different planning approaches in the Household domain and the Logistics domain.

tained by using another LLM to translate the generated PDDL domain model into natural language. Secondly, we use only two fixed examples for each domain because end users might not always be able to provide a large pool of examples, and the planner should rely on the action model information. The LLM plans, symbolic goal specifications, initial states and domain models are passed to a plan validation system (i.e., VAL) to check for unmet precondition(s) or goal condition(s). The validation results (given in PDDL) are then translated into natural language with GPT-4 and provided to the LLM planner by continuing the planning dialogue (see Appx. A.12.1 for examples). In our experiments, we limit the number of feedbacks per task to 8 due to the restricted access to GPT-4. Table 2 provides a summary of the average success rates of all approaches. Not surprisingly, the vanilla LLM planner constantly overlooks action preconditions and achieves an extremely low success rate. With the integration of validation feedback, we observe a notable improvement in plan correctness. Despite this improvement, the overall performance is still not satisfactory, as the success rate remains below

---

[3]At present, our approach only permits a fixed set of object types that are specified in the prompt. Future extensions may explore ways of enabling LLMs to create object-type hierarchy or expand the set of object types as needed.

50%. Furthermore, we have observed that GPT-4 fails to effectively utilize the feedback, often getting stuck in a loop by repeatedly generating the same plan. In some cases, it may also introduce new errors while attempting to rectify the plans.

Beyond the notion of correctness, the experiments also uncover intriguing properties of the LLM planner. In the Household domain, we intentionally introduce ordering constraints in some instructions that cannot be expressed using existing predicates (refer to Appx. A.12 for examples). Remarkably, upon manual examination of the generated plans, we observe that all LLM plans adhere to the specified ordering, despite not being entirely correct or executable. Furthermore, also in the Household domain, we observe that classical planners occasionally generate physically plausible but unconventional actions, such as placing a knife on a toaster when the knife is not being used. In contrast, the LLM planner rarely exhibits such actions, suggesting that LLMs possess knowledge of implicit human preferences. It would be meaningful to explore methods that more effectively combine the strengths of LLM planners and the correctness guarantee provided by symbolic domain models, particularly in determining which information from LLM plans should be preserved.

## 6 Conclusion

We introduce a new paradigm for leveraging LLMs in planning tasks, which involves maintaining an explicit world model instead of directly mapping user prompts to plans. This is motivated by the insight that LLMs, while incapable of the combinatorial search needed to produce correct plans, may be better suited as the source of world models. We present a complete pipeline that begins with generating high-quality PDDL models using GPT-4, then corrects the PDDL models with natural-language feedback, and finally utilizes the extracted domain models to reliably plan in multiple ways. Our experiments demonstrate that pairing LLMs with an external planner significantly outperforms existing methods when applied to two IPC domains and a household-robot domain that has more action-wise constraints than commonly used benchmarks such as ALFWorld. Apart from directions for further research that we have previously mentioned, there are several exciting opportunities for extending this work. Firstly, the complexity of our evaluation domains is still lower than that of many domains used in the classical planning literature. It remains to be seen whether LLMs can effectively scale to write PDDL models that express more intricate logic. Secondly, our framework assumes full observability, meaning that the agent must fully explore the environment to acquire object states at the beginning. It would be useful to support partial observability. Finally, our experiments assume the grounding of predicate values is done perfectly. However, it would be useful to take into account that perception can be noisy in practice.

### Acknowledgement

This research was supported by ONR grants N00014-18-1-2442, N00014-18-1-2840, N00014-19-1-2119 and N00014-23-1-2409, AFOSR grant FA9550-18-1-0067, DARPA SAIL-ON grant W911NF-19-2-0006, and a JP Morgan AI Faculty Research Grant to Kambhampati. Sreedharan was supported in part by NSF grant 2303019.

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
