# A  Appendix

# Contents

## A.1 Broader impact on using LLMs

There is a general temptation to use LLMs for a variety of tasks, including plan generation. Given the fact that LLMs cannot guarantee the generation of correct plans, this can lead to safety and security issues downstream. Our approach of teasing a domain model from LLMs, and using it in conjunction with external sound planners aims to mitigate these safety concerns. Nevertheless, given that humans are still in charge of verifying the correctness of the domain models extracted from LLMs, there is still the possibility that an incorrect or undesirable domain model is inadvertently certified correct, leading to undesirable plans and agent behaviors down the line.

Improved explainability is another advantage of extracting explicit domain models. As opposed to just directly querying LLMs for plans, generating behaviors with (intermediate) symbolic models offers additional opportunities for explanation, drawing upon existing works in explainable AI [23]. In cases where the user's understanding does not align with the transcribed model, debugging and model reconciliation techniques such as D3wa+ [52] can also be directly applied.

## A.2 Additional discussion on alternative to action-by-action PDDL construction

One alternative to our action-by-action generation could be to include descriptions of all the actions in the prompt and require the LLM to construct the entire domain model in a single dialogue. This

approach may allow the LLM to better establish a global view of all the actions. However, we do not pursue this alternative here for the following reasons: (a) the inclusion of all actions might result in a lengthy prompt, potentially exceeding the context window size of an LLM. This could pose practical issues for utilizing smaller language models (e.g., GPT-3.5-Turbo [36]) or attempting to train smaller specialized models; (b) our integration of corrective feedback relies on continuing the construction dialogue (Sec. 4.2), which necessitates a shorter initial prompt to fit within the context window; (c) our experiments indicate that the action-by-action construction approach already achieves satisfactory results

## A.3 Examples of feedback messages that capture syntax errors

Recall that we rely on the PDDL validator in VAL to identify syntax errors. However, several "simpler" syntax errors can be easily detected using simple Python scripts. In our experiments, we wrote our scripts to capture such syntax errors. Also note that since these errors can be detected at a minimal cost, the corresponding feedback messages are directly provided to the LLMs, and they are not counted in the results reported in Table 1. These "simpler" syntax errors include:

1. In this work, we only consider standard base-level PDDL. However, it's possible that the LLMs have seen various extensions of PDDL and might use them in the constructed domain models. Hence, we provide a feedback message to the LLM whenever we detect an unsupported keyword. One example feedback is: `"The precondition or effect contain the keyword 'forall' that is not supported in a standard STRIPS style model. Please express the same logic in a simplified way. You can come up with new predicates if needed (but note that you should use existing predicates as much as possible)."`

2. Newly created predicates might have the same names as existing object types, which is not allowed in PDDL. In such cases, a feedback message is provided to notify the LLM about the name clash. For instance, a message might state: `"The following predicate(s) have the same name(s) as existing object types: 1. 'smallReceptacle'. Please rename these predicates."`

3. Newly created predicates might have the same names as existing predicates, which is not allowed in PDDL. Also, LLMs often mistakenly list existing predicates under the 'New Predicates' section. In such cases, a feedback message is provided to notify the LLM about the name clash or mistake. For instance, a message might state: `"The following predicate(s) have the same name(s) as existing predicate(s): 1. (cutting-board ?z - smallReceptacle), true if the small receptacle ?z is a cutting board | existing predicate with the same name: (cutting-board ?z - householdObject), true if the object ?z is a cutting board. You should reuse existing predicates whenever possible. If you are reusing existing predicate(s), you shouldn't list them under 'New Predicates'. If existing predicates are not enough and you are devising new predicate(s), please use names that are different from existing ones. Please revise the PDDL model to fix this error."` This is the most common syntax error made by GPT-4 in our experiments.

4. The LLMs might fail to only use the object types given in the prompt. An example feedback can be: `"There is an invalid object type 'pump' for the parameter ?p."`

In our experiments, the above error types encompass the majority of syntax errors made by GPT-4. In some less-common cases, GPT-4 may have problems with predicate usage, usually caused by mis-matched object types. This kind of error can be captured by VAL and an example feedback message can be: `"There is a syntax error, the second parameter of 'object-on' should be a furnitureAppliance, but a householdObject was given. Please use the correct predicate or devise new one(s) if needed."`

## A.4 Techniques that assist users to locate errors in PDDL models

Several well-established techniques and tools are available for locating errors in PDDL models. For example, graphical tools like GIPO [49] can effectively visualize the causal dependencies of actions. However, these advanced tools or techniques are beyond the scope of this work. Here, we outline a viable solution as a starting point for users who are unfamiliar with these tools.

There are two stages in which corrective feedback can be obtained: during the construction of the PDDL model, and when the domain model is used to generate a plan. In the first stage, end users can direclty review the domain model and identify potential factual errors. Since all the predicates and parameters are accompanied by natural language descriptions, we can easily convert the PDDL model into natural language and present it to the users. This allows users to pre-screen the preconditions and effects. Note that we do not expect all factual errors to be caught in this stage, because the users may not be aware of certain constraints until they review the final plans. In the second stage, the PDDL model is used to solve downstream planning problems by following the procedure outlined in Sec. 4.3. Two possible cases may arise here: (a) no plan can be found for the given goal specification, or (b) at least one plan is found but it either gets rejected by the users or results in execution failure in the actual environment. To address the first case, we can request the users to suggest a goal-satisficing plan, which is supposed to be executable (but not necessarily optimal). We then use the generated PDDL model to "validate" the suggested plan. This allows us to find the first step in the plan that has an unsatisfied precondition(s). The models of all actions up to this step, along with the unmet precondition(s), are then converted to natural language and presented to the user for inspection. As an example, in the model of "slice an object" extracted by GPT-4 (Sec. A.6.11 in Appendix), the model requires the object to be placed on a cutting board and a furniture piece at the same time, which is not physically possible. By leveraging a user-suggested plan, we can identify the potentially erroneous model(s) and flag incorrect precondition(s). In the second case where an invalid plan is afforded by the PDDL model, there are typically missing preconditions or effects in the actions taken, both during and prior to the execution failure. The users can bring attention to these actions.

## A.5   Detailed description of the Household domain

We consider a single-arm robot model that closely resembles the SPOT robot and the Fetch Mobile Manipulator. Consequently, the robot is incapable of grasping multiple objects simultaneously or executing manipulation actions while holding irrelevant items (e.g., opening a fridge door while holding a mug). We also ensure the constraints align with real-world robotic capabilities. For example, we recognize that robot arms may be significantly less flexible than human arms, and therefore, we require certain manipulation tasks to be performed on furniture pieces with open and flexible surfaces (e.g., the robot can only pick up food items from a lunch box when the lunch box is placed on a kitchen countertop instead of inside a fridge). The list of actions and their descriptions can be found in Appendix starting from Sec. A.6.2. The PDDL-construction prompt for this domain includes a general description of the domain, which outlines the tasks to be performed by the robot, the types of objects involved, and details of the robot's morphology. An example of a complete prompt can be found in Fig. A.6.1.

## A.6   Household: Constructing PDDL Models

The AI agent here is a household robot that can navigate to various large and normally immovable furniture pieces or appliances in the house to carry out household tasks. Note that the robot has only one gripper, so (a) it can only hold one object; (b) it shouldn't hold any other irrelevant objects in its gripper while performing some manipulation tasks (e.g., opening a drawer or closing a window); (c) operations on small household items should be carried out on furniture with a flat surface to get enough space for manipulation. There are three major types of objects in this domain: robot, furnitureAppliance, and householdObject. The object type furnitureAppliance covers large and normally immovable furniture pieces or appliances, such as stove burners, side tables, dining tables, drawer, cabinets, or microwaves. The object type householdObject covers all other small household items, such as handheld vacuum cleaners, cloth, apples, bananas, and small receptacles like bowls and lunch boxes. There is a subtype of householdObject called smallReceptacle that covers small receptacles like bowls, lunch boxes, plates, etc. In this domain, the locations of the robot and small household items (e.g., apples, oranges, bowls, lunch boxes or lamps) are determined by large and normally immovable furniture pieces or appliances.

### A.6.1 An example prompt for constructing PDDL models of the action "close a small receptacle"

---

**An example prompt for constructing PDDL models of the action "close a small receptacle"**

Instructions for the PDDL generation task
You are defining the preconditions and effects (represented in PDDL format) of an AI agent's
↪ actions. Information about the AI agent will be provided in the domain description. Note that
↪ individual conditions in preconditions and effects should be listed separately. For example,
↪ ''object_1 is washed and heated'' should be considered as two separate conditions ''object_1 is
↪ washed'' and ''object_1 is heated''. Also, in PDDL, two predicates cannot have the same name even
↪ if they have different parameters. Each predicate in PDDL must have a unique name, and its
↪ parameters must be explicitly defined in the predicate definition. It is recommended to define
↪ predicate names in an intuitive and readable way.
One or two examples from other domains for illustrating the input and output formats
Here are two examples from the classical BlocksWorld domain for demonstrating the output format.

Domain information: BlocksWorld is a planning domain in artificial intelligence. The AI agent here
↪ is a mechanical robot arm that can pick and place the blocks. Only one block may be moved at a
↪ time: it may either be placed on the table or placed atop another block. Because of this, any
↪ blocks that are, at a given time, under another block cannot be moved. There is only one type
↪ of object in this domain, and that is the block.

Example 1
Action: This action enables the robot to put a block onto the table. For example, the robot puts
↪ block_1 onto the table.

You can create and define new predicates, but you may also reuse the following predicates:
No predicate has been defined yet

Parameters:
1. ?x - block: the block to put down

Preconditions:
```
(and
    (robot-holding ?x)
)
```

Effects:
```
(and
    (not (robot-holding ?x))
    (block-clear ?x)
    (robot-hand-empty)
    (block-on-table ?x)
)
```

New Predicates:
1. (robot-holding ?x - block): true if the robot arm is holding the block ?x
2. (block-clear ?x - block): true if the block ?x is not under any another block
3. (robot-hand-empty): true if the robot arm is not holding any block
4. (block-on-table ?x - block): true if the block ?x is placed on the table

Example 2
Action: This action enables the robot to pick up a block on the table.

You can create and define new predicates, but you may also reuse the following predicates:
1. (robot-holding ?x - block): true if the robot arm is holding the block ?x
2. (block-clear ?x - block): true if the block ?x is not under any another block
3. (robot-hand-empty): true if the robot arm is not holding any block
4. (block-on-table ?x - block): true if the block ?x is placed on the table

Parameters:
1. ?x - block: the block to pick up

Preconditions:
```
(and
    (block-clear ?x)
    (block-on-table ?x)
    (robot-hand-empty)
)
```

```
Effects:
```
(and
    (not (block-on-table ?x))
    (not (block-clear ?x))
    (not (robot-hand-empty))
    (robot-holding ?x)
)
```

New Predicates:
No newly defined predicate

Here is the task.
```

```
Domain information: The AI agent here is a household robot that can navigate to various large and
↪    normally immovable furniture pieces or appliances in the house to carry out household tasks.
↪    Note that the robot has only one gripper, so (a) it can only hold one object; (b) it shouldn't
↪    hold any other irrelevant objects in its gripper while performing some manipulation tasks
↪    (e.g., opening a drawer or closing a window); (c) operations on small household items should
↪    be carried out on furniture with a flat surface to get enough space for manipulation. There
↪    are three types of objects in this domain: robot, furnitureAppliance, and householdObject. The
↪    object type furnitureAppliance covers large and normally immovable furniture pieces or
↪    appliances, such as stove burners, side tables, dining tables, drawer, cabinets, or microwaves.
↪    The object type householdObject covers all other small household items, such as handheld
↪    vacuum cleaners, cloth, apples, bananas, and small receptacles like bowls and lunch boxes. In
↪    this domain, the locations of the robot and small household items (e.g., apples, oranges,
↪    bowls, lunch boxes or lamps) are determined by large and normally immovable furniture pieces
↪    or appliances.
```

```
Action: This action enables the robot to close a small receptacle receptacle_1 that is openable
↪    (e.g. small storage boxes or lunch boxes with lids). For example, the robot closes lunch_box_2,
↪    or the robot closes storage_box_1.
```

```
You can create and define new predicates, but you may also reuse the following predicates:
1. (robot-at ?x - furnitureAppliance): true if the robot is at the furniture or appliance ?x
2. (object-on ?x - householdObject ?y - furnitureAppliance): true if the object ?x is on or in the
↪    furniture or appliance ?y
3. (pickupable ?x - householdObject): true if the object ?x can be picked up by the robot
4. (closed-receptacle ?x - smallReceptacle): true if the small receptacle ?x is closed
...

Parameters:
```

## A.6.2 Navigate to a furniture piece or an appliance

### Action description

```
This action enables the robot to navigate from one normally immovable piece of furniture to
↪    another (e.g., dining tables, side tables, cabinets, and sinks) or an appliance (e.g.,
↪    refrigerators, coffee makers, microwaves, and washers).
```

### GPT-4: Navigate to a furniture piece or an appliance

```
Parameters:
1. ?x - furnitureAppliance: the furniture or appliance the robot is currently at
2. ?y - furnitureAppliance: the furniture or appliance the robot wants to navigate to

Preconditions:
(and
    (robot-at ?x)
    (not (= ?x ?y))
)

Effects:
(and
    (not (robot-at ?x))
    (robot-at ?y)
)
```

### A.6.3 Pick up an object on or in a furniture piece or an appliance

---

**Action description**

This action enables the robot to pick up an object object_1 in/on a large piece of furniture (e.g.,
↪   dining tables, drawers, cabinets) or an appliance (e.g., dishwashers and refrigerators). For
↪   example, the robot pick up coffee_mug_1 on dining_table_1, the robot pick up fork_1 in
↪   drawer_1, or the robot pick up apple_1 in refrigerator_1.

---

**Additional information from the user**

(a) This action is only applicable for objects that are pickupable by the robot.
(b) The furniture piece or appliance should be opened if it is openable.
(c) The object to pick up should not be stacked on top of other household items.

---

**GPT-4: Pick up an object on or in a furniture piece or an appliance**

```
Parameters:
1. ?x - householdObject: the object to pick up
2. ?y - furnitureAppliance: the furniture or appliance where the object is located

Preconditions:
(and
    (robot-at ?y)
    (object-on ?x ?y)
    (pickupable ?x)
    Inaccurate precondition: "?x is not stacked on top of other object" should
        only apply when ?x is a stackable object
    (not (object-stacked ?x))
    (robot-hand-empty)
    (or
        (not (openable ?y))
        (opened ?y)
    )
    Missing precondition: there should be no other object stacked on top of
        ?x if ?x is a stackable object
)

Effects:
(and
    (not (object-on ?x ?y))
    (robot-holding ?x)
    (not (robot-hand-empty))
)
```

### A.6.4 Put an object on or in a furniture piece or an appliance

---

**Action description**

This action enables the robot to put an object object_1 in/on a large piece of furniture (e.g.,
↪   dining tables, drawers, cabinets) or an appliance (e.g., dishwashers and refrigerators). For
↪   example, the robot puts coffee_mug_1 on dining_table_1, the robot puts fork_1 in drawer_1, or
↪   the robot puts apple_1 in refrigerator_1.

---

**Additional information from the user**

(a) The furniture piece or appliance should be opened if it is openable.

---

**GPT-4: Put an object on or in a furniture piece or an appliance**

```
Parameters:
1. ?x - householdObject: the object to put in/on the furniture or appliance
2. ?y - furnitureAppliance: the furniture or appliance to put the object in/on

Preconditions:
(and
    (robot-at ?y)
    (robot-holding ?x)
    (pickupable ?x)
```

```
        Unnecessary precondition "no other object stacked on object ?x"
        (object-clear ?x)
        (or
            (not (openable ?y))
            (opened ?y)
        )
    )

    Effects:
    (and
        (not (robot-holding ?x))
        (robot-hand-empty)
        (object-on ?x ?y)
        Syntax error, there is an illegal keyword "if"
        (if (openable ?y) (closed ?y))
    )
```

### A.6.5   Stack Objects

**Action description**

```
This action enables the robot to stack one object object_1 on top of another object object_2 that
↪    is on the same piece of furniture. For example, the robot stacks book_1 on top of book_2, or
↪    the robot stacks lunch_box_1 on top of lunch_box_2.
```

**Additional information from the user**

```
(a) The furniture piece has to have an open and flat surface.
(b) Note that both object_1 and object_2 must be stackable objects.
(c) You may assume the robot is holding object_1 at the beginning.
(d) Also, there should be no other object on top of object_2.
```

**GPT-4: Stack objects**

```
Parameters:
1. ?x - householdObject: the object to stack on top of another object
2. ?y - householdObject: the object on which ?x will be stacked
3. ?z - furnitureAppliance: the furniture or appliance on which both objects are placed

Preconditions:
(and
    (robot-holding ?x)
    (object-on ?y ?z)
    (object-clear ?y)
    (flat-surface ?z)
    Incorrect precondition: should be stackable
    (pickupable ?x)
    Incorrect precondition: should be stackable
    (pickupable ?y)
    (robot-at ?z)
)

Effects:
(and
    (not (robot-holding ?x))
    (not (object-clear ?y))
    (robot-hand-empty)
    (object-on ?x ?z)
    (object-stacked ?x)
    Missing effect: object ?x should be stacked on top of object ?y
)
```

### A.6.6   Unstack Objects

**Action description**

```
This action enables the robot to unstack one object object_1 that is on top of another object
↪    object_2. For example, the robot stacks book_1 on top of book_2, or the robot stacks
↪    lunch_box_1 on top of lunch_box_2. The robot will hold the object after unstacking it.
```

```
Parameters:
1. ?x - householdObject: the object to unstack from another object
2. ?y - householdObject: the object that ?x is stacked on
3. ?z - furnitureAppliance: the furniture or appliance where the objects are stacked

Preconditions:
(and
    (robot-at ?z)
    (object-on ?x ?z)
    (object-on ?y ?z)
    (pickupable ?x)
    (object-stacked ?x)
    (object-clear ?x)
    (robot-hand-empty)
    Missing precondition: object ?x should be stacked on top of object ?y
)

Effects:
(and
    (not (object-on ?x ?z))
    (not (object-stacked ?x))
    Incorrect effect, this action shouldn't change whether or not object ?x
        is under any another household object
    (not (object-clear ?x))
    (not (robot-hand-empty))
    (robot-holding ?x)
    (object-clear ?y)
    Missing effect: object ?x is no longer stacked on top of object ?y
)
```

### A.6.7 Open a furniture piece or an appliance

Action description

```
This action enables the robot to open a large piece of furniture (e.g., cabinets and drawers) or
↪  an appliance (e.g., dishwashers and refrigerators) that is openable. For example, the robot
↪  opens cabinet_1, or the robot opens drawer_2.
```

GPT-4: Open a furniture piece or an appliance

```
Parameters:
1. ?y - furnitureAppliance: the furniture or appliance to be opened

Preconditions:
(and
    (robot-at ?y)
    (openable ?y)
    (closed ?y)
    (robot-hand-empty)
)

Effects:
(and
    (not (closed ?y))
    (opened ?y)
)
```

### A.6.8 Close a furniture piece or an appliance

Action description

```
This action enables the robot to close a large piece of furniture (e.g., cabinets and drawers) or
↪  an appliance (e.g., dishwashers and refrigerators) that is openable. For example, the robot
↪  closes cabinet_1, or the robot closes drawer_2.
```

### GPT-4: Close a furniture piece or an appliance

```
Parameters:
1. ?y - furnitureAppliance: the furniture or appliance to close

Preconditions:
(and
    (robot-at ?y)
    (openable ?y)
    (opened ?y)
    (robot-hand-empty)
)

Effects:
(and
    (not (opened ?y))
    (closed ?y)
)
```

### A.6.9   Toggle a small appliance on

### Action description

```
This action enables the robot to toggle a small appliances (like humidifiers and light bulbs) to
↪  switch them on. For example, the robot toggles humidifier_2 on, or the robot toggle
↪  light_bulb_1 on.
```

### GPT-4: Toggle a small appliance on

```
Parameters:
1. ?x - householdObject: the small appliance to toggle on
2. ?y - furnitureAppliance: the furniture or appliance where the small appliance is located

Preconditions:
(and
    (robot-at ?y)
    (object-on ?x ?y)
    (robot-hand-empty)
    (not (appliance-on ?x))
    Missing precondition: the appliance to toggle should be a toggleable appliance
)

Effects:
(and
    (appliance-on ?x)
)
```

### A.6.10   Toggle a small appliance off

### Action description

```
This action enables the robot to toggle a small appliances (like humidifiers and light bulbs) to
↪  switch them off. For example, the robot toggles humidifier_2 off, or the robot toggle
↪  light_bulb_1 off.
```

### GPT4: Toggle a small appliance off

```
Parameters:
1. ?r - robot: the robot performing the action
2. ?o - householdObject: the small appliance to be toggled off
3. ?f - furnitureAppliance: the furniture or appliance where the small appliance is located

Preconditions:
(and
    (robot-at ?r ?f)
    (object-in-on-furniture ?o ?f)
    (robot-gripper-empty ?r)
    (appliance-on ?o)
)

Effects:
```

```
(and
    (not (appliance-on ?o))
)
```

### A.6.11 Slice an object

**Action description**

```
This action enables the robot to slice objects (like fruits and vegetables) with a knife. For
↪    example, the robot slices orange_1 with knife_2, the robot slices potato_3 with knife_1.
```

**Additional information from the user**

```
(a) Note that the object to slice needs to be placed on a cutting board. A cutting board can be
↪    viewed as a special kind of receptacle.
(b) You may assume the robot is holding the knife in its gripper at the beginning.
(c) The object to slice should be sliceable.
(d) The furniture piece needs to have an open and flat surface to allow enough space for
↪    manipulation (e.g., countertops or tables).
```

**GPT-4: Slice an object**

```
Parameters:
1. ?x - householdObject: the object to slice
2. ?k - householdObject: the knife to use for slicing
3. ?y - furnitureAppliance: the furniture or appliance with a flat surface for manipulation
Incorrect object type, a cutting board should be viewed as
    a small receptacle as mentioned in the prompt
4. ?z - householdObject: the cutting board on which the object is placed

Preconditions:
(and
    (robot-at ?y)
    Incorrect precondition: ?x should be on the cutting board not directly on the furniture ?y
    (object-on ?x ?y)
    (object-on ?z ?y)
    (robot-holding ?k)
    (sliceable ?x)
    (cutting-board ?z)
    (knife ?k)
    (flat-surface ?y)
    (object-whole ?x)
)

Effects:
(and
    (not (object-whole ?x))
    (object-sliced ?x)
)
```

### A.6.12 Heat food with a microwave

**Action description**

```
This action enables the robot to start a microwave and heat up the food inside. For example, the
↪    robot heats pizza_1 with microwave_2, the robot heats pasta_2 with microwave_1.
```

**Additional information from the user**

```
(a) The food to heat should be placed in a small receptacle (e.g., a plate or a bowl).
(b) The robot also needs to close the door of the microwave before taking this action.
(c) Note that the food is no longer pickupable after it has been heated.
```

**GPT-4: Heat food with a microwave**

```
Parameters:
1. ?x - householdObject: the food to heat up
```

```
    2. ?y - furnitureAppliance: the microwave to use
    3. ?z - smallReceptacle: the small receptacle containing the food

    Preconditions:
    (and
        (robot-at ?y)
        Incorrect precondition: the object should be in the receptacle not directly in the microwave
        (object-on ?x ?y)
        (object-in ?x ?z)
        (is-small-receptacle ?z)
        (object-on ?z ?y)
        Incorrect precondition: ?x needs not be pickupable
        (pickupable ?x)
        (closed ?y)
        (not (food-heated ?x))
        Missing precondition: ?y should be a microwave
    )

    Effects:
    (and
        (not (pickupable ?x))
        (food-heated ?x)
    )
```

### A.6.13 Heat food with pan

**Action description**

```
This action enables the robot to heat food which is heatable with a pan. For example, the robot
↪  heats pizza_1 with pan_2, the robot heats pasta_2 with pan_1.
```

**Additional information from the user**

```
(a) Note that the food should be placed on the pan, and the pan needs to be placed on a stove
↪  burner before executing this action.
(b) A stove burner can be viewed as an immovable appliance, and a pan can be viewed as a small
↪  receptacle.
(c) Note that the food is no longer pickupable after it has been heated.
```

**GPT-4: Heat food with pan**

```
Parameters:
1. ?f - householdObject: the food to be heated
Inaccurate object type, a pan ?p can be viewed as a small receptacle
2. ?p - householdObject: the pan to be used for heating the food
3. ?s - furnitureAppliance: the stove burner on which the pan is placed

Preconditions:
(and
    (robot-at ?s)
    Syntax error: the second parameter of `object-on` should
        be a furnitureAppliance, but a householdObject was given
    (object-on ?f ?p)
    (object-on ?p ?s)
    Incorrect precondition: ?f needs not be pickupable
    (pickupable ?f)
    (is-small-receptacle ?p)
    Incorrect precondition: a stove or burner ?s doesn't need to have a flat surface
    (flat-surface ?s)
    (robot-hand-empty)
    Missing precondition: ?p has to be a pan
    Missing precondition: ?s has to be a stove burner
)

Effects:
(and
    (not (pickupable ?f))
    (food-heated-pan ?f)
)
```

### A.6.14 Transfer food from one small receptacle to another

This action enables the robot to transfer food food_1 from one small receptacle receptacle_1 to
↪    another small receptacle receptacle_2. This skill is usually used to transfer food that can
↪    not be easily picked up by the robot. For example, the robot can pour the milk from milk_box_1
↪    to mug_1, or the robot can transfer heated pizza_1 from pan_2 to plate_1.

**GPT-4: Transfer food from one small receptacle to another**

```
Parameters:
1. ?food - householdObject: the food to be transferred
2. ?receptacle1 - smallReceptacle: the small receptacle from which the food will be transferred
3. ?receptacle2 - smallReceptacle: the small receptacle to which the food will be transferred
4. ?furniture - furnitureAppliance: the furniture piece with a flat surface where the transfer
↪    will take place

Preconditions:
(and
    (robot-at ?furniture)
    (object-on ?receptacle1 ?furniture)
    (object-on ?receptacle2 ?furniture)
    (object-in ?food ?receptacle1)
    (is-small-receptacle ?receptacle1)
    (is-small-receptacle ?receptacle2)
    (flat-surface ?furniture)
    (robot-hand-empty)
    Inaccurate precondition: "?receptacle1 is not stacked on other objects" should
        only apply if ?receptacle1 is a stackable object
    (not (object-stacked ?receptacle1))
    Inaccurate precondition: "?receptacle2 is not stacked on other objects" should
        only apply if ?receptacle2 is a stackable object
    (not (object-stacked ?receptacle2))
    (or (not (openable-receptacle ?receptacle1)) (opened-receptacle ?receptacle1))
    (or (not (openable-receptacle ?receptacle2)) (opened-receptacle ?receptacle2))
    Missing precondition: there should be no other object stacked
        on top of ?receptacle1 if ?receptacle1 is stackable
    Missing precondition: there should be no other object stacked
        on top of ?receptacle2 if ?receptacle2 is stackable
)

Effects:
(and
    (not (object-in ?food ?receptacle1))
    (object-in ?food ?receptacle2)
)
```

### A.6.15 Put an object onto or into a small receptacle like a bowl and plate

```
Parameters:
1. ?x - householdObject: the object to put into/onto the small receptacle
2. ?y - furnitureAppliance: the furniture or appliance with a flat surface for manipulation
3. ?z - smallReceptacle: the small receptacle to put the object into/onto

Preconditions:
(and
    (robot-at ?y)
    Inaccurate precondition: object ?x can't be on the furnitureAppliance ?y
        when ?x is held by the robot
    (object-on ?x ?y)
    (pickupable ?x)
    Inaccurate precondition: this should only apply if ?z is stackable
    (not (object-stacked ?z))
    (flat-surface ?y)
    (robot-holding ?x)
    (is-small-receptacle ?z)
    (object-on ?z ?y)
    Inaccurate precondition:  this should only apply if ?z is stackable
    (object-clear ?z)
    Unnecessary precondition: the robot holding the object ?x already
        implies ?x is not in the receptacle ?z
    (not (object-in ?x ?z))
    (or
        (not (openable-receptacle ?z))
        (opened-receptacle ?z)
    )
)

Effects:
(and
    (not (robot-holding ?x))
    (not (object-on ?x ?y))
    (robot-hand-empty)
    (object-in ?x ?z)
)
```

### A.6.16   Pick up an object on or in a small receptacle like a bowl and plate

**Action description**

```
This action enables the robot to pick up an object object_1 in some small receptacle receptacle_1
↪  (e.g. storage boxes, lunch boxes, bowls, plates). For example, the robot picks up apple_1 in
↪  blow_1, or the robot picks up apple_2 in lunch_box_2.
```

**Additional information from the user**

```
(a) The furniture piece needs to have an open and flat surface to allow enough space for
↪  manipulation (e.g., countertops or tables).
(b) The receptacle should be opened if it is openable.
```

```
Parameters:
1. ?x - householdObject: the object to pick up
2. ?y - furnitureAppliance: the furniture or appliance where the object is located
3. ?z - smallReceptacle: the small receptacle containing the object

Preconditions:
(and
    (robot-at ?y)
    Incorrect precondition: the object ?x is in the receptacle ?z
        so ?x is not directly on the furniture or appliance ?y
    (object-on ?x ?y)
    (pickupable ?x)
    (object-in ?x ?z)
    (is-small-receptacle ?z)
    Inaccurate precondition: "?z is not stacked on other objects" should
        only apply if ?z is a stackable object
    (not (object-stacked ?z))
```

```
        (flat-surface ?y)
        (robot-hand-empty)
        Incorrect precondition: ?x can't be stacked inside a receptacle so the precondition `?x is not
            under other household object` is unnecessary
        (object-clear ?x)
        Incorrect precondition: ?z can be an openable receptacle
        (not (openable-receptacle ?z))
        (or
            Syntax error, the first parameter of `openable` should
                be a furnitureAppliance but here a smallReceptacle is given
            (not (openable ?z))
            (opened-receptacle ?z)
        )
        Missing precondition: there should be no other object stacked on top of ?z if ?z is stackable
)

Effects:
(and
    (not (object-on ?x ?y))
    (not (object-in ?x ?z))
    (not (robot-hand-empty))
    (robot-holding ?x)
)
```

### A.6.17 Open a small receptacle such as a lunch box with a lid

#### Action description

```
This action enables the robot to open a small receptacle receptacle_1 (e.g. small storage boxes or
↪   lunch boxes with lids). For example, the robot opens lunch_box_2, or the robot opens
↪   storage_box_1.
```

#### Additional information from the user

```
(a) This action is only applicable for receptacles that are openable.
(b) The receptacle needs to be placed on a furniture piece that has an open and flat surface to
↪   allow enough space for manipulation (e.g., countertops or tables).
(c) The receptacle should not be stacked on top of other objects if it is stackable.
```

#### GPT-4: Open a small receptacle such as a lunch box with a lid

```
Parameters:
1. ?x - smallReceptacle: the small receptacle to be opened
2. ?y - furnitureAppliance: the furniture or appliance on which the small receptacle is placed

Preconditions:
(and
    (robot-at ?y)
    (object-on ?x ?y)
    (openable-receptacle ?x)
    (flat-surface ?y)
    (robot-hand-empty)
    Inaccurate precondition: should only apply if ?x is stackable
    (not (object-stacked ?x))
    (closed-receptacle ?x)
    Missing precondition: there should be no other object stacked on top of ?x
)

Effects:
(and
    (not (closed-receptacle ?x))
    (opened-receptacle ?x)
)
```

## A.6.18 Close a small receptacle such as a lunch box with a lid

**Action description**

```
This action enables the robot to close a small receptacle receptacle_1 that is openable (e.g.
↪   small storage boxes or lunch boxes with lids). For example, the robot closes lunch_box_2, or
↪   the robot closes storage_box_1.
```

**Additional information from the user**

```
(a) This action is only applicable for receptacles that are openable.
(b) The receptacle needs to be placed on a furniture piece that has an open and flat surface to
↪   allow enough space for manipulation (e.g., countertops or tables).
(c) The receptacle should not be stacked on top of other objects if it is stackable.
```

**GPT-4: Close a small receptacle such as a lunch box with a lid**

```
Parameters:
1. ?x - smallReceptacle: the small receptacle to close
2. ?y - furnitureAppliance: the furniture or appliance on which the small receptacle is placed

Preconditions:
(and
    (robot-at ?y)
    (object-on ?x ?y)
    (openable-receptacle ?x)
    (opened-receptacle ?x)
    (flat-surface ?y)
    (robot-hand-empty)
    Inaccurate precondition: "?x is not stacked on other objects" should
        only apply if ?x is a stackable object
    (not (object-stacked ?x))
    Missing precondition: there should be no other object stacked on top of ?x
)

Effects:
(and
    (not (opened-receptacle ?x))
    (closed-receptacle ?x)
)
```

## A.6.19 Mash food with a blender

**Action description**

```
This action enables the robot to use a blender to mash some food in it. For example, the robot
↪   gets mashed apple_2 with blender_1, or the robot gets mashed banana_1 with blender_2.
```

**Additional information from the user**

```
(a) The food needs to be sliced beforehand and the food should be placed inside the blender.
(b) Note that the food remains in the blender after this action is performed.
(c) You may also assume the blender is turned off before and after mashing the food.
```

**GPT-4: Mash food with a blender**

```
Parameters:
Incorrect object type: a blender ?b should be a smallReceptacle
1. ?b - householdObject: the blender to be used for mashing the food
2. ?f - householdObject: the food to be mashed
3. ?y - furnitureAppliance: the furniture or appliance where the blender is located

Preconditions:
(and
    (robot-at ?y)
    (object-on ?b ?y)
    Incorrect precondition: the food to mash should be in the blender ?b
        rather than directly on the furniture piece ?y
    (object-on ?f ?y)
```

```
        (object-in ?f ?b)
        (object-sliced ?f)
        (is-small-receptacle ?b)
        Incorrect precondition: as mentioned in the prompt, the blender
            should be turned off at the beginning
        (appliance-on ?b)
        (robot-hand-empty)
        Missing precondition: ?b should be a blender
    )

    Effects:
    (and
        (not (appliance-on ?b))
        (object-mashed ?f)
        Missing effect: ?f is no longer pickupable
    )
```

### A.6.20   Wash an object

**Action Description**

This action enables the robot to wash an object (e.g., fruits and cloths) in a sink or basin. For
↪   example, the robot washed apple_1, or the robot washed romaine_lettuce_1.

**Additional information from the user**

(a) The object has to be something washable.
(b) The robot should hold the object when washing it.

**GPT-4: Wash an object**

```
Parameters:
1. ?x - householdObject: the object to be washed
2. ?y - furnitureAppliance: the sink or basin where the object will be washed

Preconditions:
(and
    (robot-at ?y)
    (washable ?x)
    (robot-holding ?x)
    Missing precondition: ?y should be a sink or basin
)

Effects:
(and
    (not (object-dirty ?x))
    (object-washed ?x)
)
```

### A.6.21   Wipe a surface

**Action description**

This action enables the robot to wipe and clean the surface of a piece of furniture or an
↪   appliance, such as a dining table, a mirror, a sink, or a bathtub, with a cloth. For example,
↪   the robot wipes dining_table_1 with cloth_1, the robot wipes mirror_1 with cloth_2.

**Additional information from the user**

(a) You may assume the robot is holding the cloth before executing this action.
(b) The cloth will be dirty after executing the action.
(c) The robot should also use a clean cloth.

```
Parameters:
1. ?x - furnitureAppliance: the furniture or appliance to be wiped and cleaned
2. ?c - householdObject: the cloth to be used for wiping and cleaning

Preconditions:
(and
    (robot-at ?x)
    (robot-holding ?c)
    (washable ?c)
    (object-washed ?c)
    Incorrect precondition: ?x doesn't need to have a flat surface
    (flat-surface ?x)
    Missing precondition: ?c should be a cloth
)

Effects:
(and
    (not (object-washed ?c))
    (object-dirty ?c)
    (furniture-cleaned ?x)
)
```

## A.6.22 Vacuum a carpet

### Action description

```
This action enables the robot to vacuum a carpet with a handheld vacuum cleaner. For example, the
↪  robot uses vacuum_cleaner_1 to vacuum carpet_2, the robot uses vacuum_cleaner_2 to vacuum
↪  carpet_1.
```

### Additional information from the user

```
(a) You need to make sure the dust bin of the vacuum cleaner is not full before executing this
↪  action.
(b) You may assume the robot is holding the vacuum cleaner at the beginning.
(c) The dust bin of the vacuum cleaner will be full of dust.
```

### GPT4: Vacuum a carpet

```
Parameters:
1. ?v - householdObject: the handheld vacuum cleaner to use
2. ?c - furnitureAppliance: the carpet to vacuum

Preconditions:
(and
    (robot-holding ?v)
    (robot-at ?c)
    (pickupable ?v)
    (not (dustbin-full ?v))
    (not (carpet-clean ?c))
    Missing precondition: ?v should be a handheld vacuum cleaner
    Missing precondition: ?c should be a carpet
)

Effects:
(and
    (carpet-clean ?c)
    (dustbin-full ?v)
)
```

## A.6.23 Empty a vacuum cleaner

### Action description

```
This action enables the robot to empty a vacuum cleaner's dust bin and dump the dust into a trash
↪  can. For example, the robot empties vacuum_cleaner_1 and dumps the dust into trash_can_2, or
↪  the robot empties vacuum_cleaner_2 and dumps the dust into trash_can 1.
```

## Additional information from the user

(a) Note that the robot should hold the vacuum cleaner and stand by the trash can before executing
↪  this action.
(b) After executing this action, the robot is still holding the vacuum cleaner.
(c) The trash can should be opened if it's openable.
(d) The dust bin will be empty after executing this action.

## GPT-4: Empty a vacuum cleaner

```
Parameters:
1. ?v - householdObject: the vacuum cleaner with a full dust bin
2. ?t - furnitureAppliance: the trash can where the dust will be dumped

Preconditions:
(and
    (robot-at ?t)
    (robot-holding ?v)
    (dustbin-full ?v)
    (pickupable ?v)
    Inaccurate precondition: ?t should be open only if it is openable
    (openable ?t)
    (opened ?t)
    Missing precondition: ?t should be a trash can
    Missing precondition: ?v should be a vacuum cleaner
)

Effects:
(and
    (not (dustbin-full ?v))
)
```

### A.6.24 Examples of PDDL action models constructed by GPT-3.5-Turbo

## GPT-3.5-Turbo: Navigate to a furniture piece or an appliance

```
Parameters:
1. ?f1 - furnitureAppliance: the furniture/appliance to navigate from
2. ?f2 - furnitureAppliance: the furniture/appliance to navigate to

Preconditions:
(and
    (robot-at ?f1)
    (reachable ?f1 ?f2)
    Irrelevant preconditions
    (not (robot-holding ?o))
    Irrelevant preconditions
    (forall (?o - householdObject)
        (not (object-on ?o ?f1))
    )
)

Effects:
(and
    (robot-at ?f2)
    Missing effect: (not (robot-at ?f1))
)
```

## GPT-3.5-Turbo: Pick up an object on or in a furniture piece or an appliance

```
Parameters:
1. ?o - householdObject: the object to pick up
2. ?f - furnitureAppliance: the furniture/appliance on which the object is located

Preconditions:
(and
    (robot-at ?f)
    Incorrect precondition and unknown parameter: ?f1
    (reachable ?f1 ?f)
    (object-on ?o ?f)
    (object-pickupable ?o)
    Inaccurate precondition: "?x is not stacked on top of other object" should
```

```
        only apply when ?x is a stackable object
    (not (object-stacked ?o))
    Inaccurate precondition: the gripper should be empty
    (not (robot-holding ?o))
    (or (not (furniture-openable ?f)) (furniture-open ?f))
    Missing precondition: there should be no other object stacked
        on top of ?x if ?x is a stackable object
)

Effects:
(and
    (not (object-on ?o ?f))
    Incorrect effect and unknown parameter: ?f1
    (not (object-in ?o ?c))
    Incorrect effect and unknown parameter: ?o2
    (not (object-stacked-on ?o ?o2))
    Incorrect effect
    (not (object-washed ?o))
    (robot-holding ?o)
)
```

```
1. ?object_1 - householdObject: the object to be stacked on top of another object
2. ?object_2 - householdObject: the object on which ?object_1 will be stacked
Missing parameter: ?f, the flat furniture piece

Preconditions:
(and
    Syntax error
    (robot-at ?f - furnitureAppliance)
    Syntax error and incorrect precondition
    (reachable ?f - furnitureAppliance ?f2 - furnitureAppliance)
    Incorrect precondition: should be stackable
    (object-pickupable ?object_1)
    Syntax error
    (object-stacked-on ?object_2 ?o - householdObject)
    Unnecessary precondition
    (not (object-stacked ?object_1))
    Unknown parameter
    (not (object-stacked-on ?o ?object_1))
    (object-on ?object_2 ?f)
    Incorrect precondition: the furniture piece should be flat not openable
    (furniture-openable ?f)
    Incorrect precondition
    (furniture-open ?f)
    (robot-holding ?object_1)
    Missing precondition: there should be no other objects on top of ?object_2
    Missing precondition: ?object_2 should be stackable
)

Effects:
(and
    (not (robot-holding ?object_1))
    (object-stacked-on ?object_1 ?object_2)
    Incorrect effect
    (not (object-on ?object_1 ?f))
)
```

### A.6.25 The initial set of predicates extracted by GPT-4

#### The initial set of predicates extracted by GPT-4

```
1. (robot-at ?x - furnitureAppliance): true if the robot is at the furniture or appliance ?x
2. (object-on ?x - householdObject ?y - furnitureAppliance): true if the object ?x is on or in the
↪    furniture or appliance ?y
3. (pickupable ?x - householdObject): true if the object ?x can be picked up by the robot
4. (object-stacked ?x - householdObject): true if the object ?x is stacked on top of other
↪    household items
5. (openable ?y - furnitureAppliance): true if the furniture or appliance ?y can be opened
6. (opened ?y - furnitureAppliance): true if the furniture or appliance ?y is opened
7. (robot-holding ?x - householdObject): true if the robot is holding the object ?x
8. (robot-hand-empty): true if the robot's gripper is empty
```

```
9. (object-clear ?x - householdObject): true if the object ?x is not under any another household
↪ object
10. (appliance-on ?x - householdObject): true if the small appliance ?x is switched on
11. (sliceable ?x - householdObject): true if the object ?x can be sliced
12. (cutting-board ?z - householdObject): true if the object ?z is a cutting board
13. (knife ?k - householdObject): true if the object ?k is a knife
14. (flat-surface ?y - furnitureAppliance): true if the furniture or appliance ?y has a flat
↪ surface for manipulation
15. (object-whole ?x - householdObject): true if the object ?x is whole and not sliced
16. (object-sliced ?x - householdObject): true if the object ?x has been sliced
17. (food-heated ?x - householdObject): true if the food ?x has been heated in the microwave
18. (food-heated-pan ?x - householdObject): true if the food ?x has been heated with a pan
19. (is-small-receptacle ?x - householdObject): true if the object ?x is a small receptacle like
↪ bowls, lunch boxes, plates, etc.
20. (object-in ?x - householdObject ?z - smallReceptacle): true if the object ?x is in the small
↪ receptacle ?z
21. (openable-receptacle ?x - smallReceptacle): true if the small receptacle ?x can be opened
22. (opened-receptacle ?x - smallReceptacle): true if the small receptacle ?x is opened
23. (object-mashed ?x - householdObject): true if the object ?x has been mashed in the blender
24. (washable ?x - householdObject): true if the object ?x can be washed
25. (object-washed ?x - householdObject): true if the object ?x has been washed
26. (object-dirty ?x - householdObject): true if the object ?x is dirty
27. (dustbin-full ?v - householdObject): true if the dust bin of the vacuum cleaner ?v is full
28. (carpet-clean ?c - furnitureAppliance): true if the carpet ?c is clean
29. (closed ?y - furnitureAppliance): true if the furniture or appliance ?y is closed
30. (closed-receptacle ?x - smallReceptacle): true if the small receptacle ?x is closed
31. (furniture-cleaned ?x - furnitureAppliance): true if the furniture or appliance ?x has been
↪ wiped and cleaned
```

## A.7 Household: Correcting PDDL Models

### A.7.1 Heat food with a pan

---

**GPT-4: Heat food with a pan**

User:
```
...
The original prompt for PDDL construction
...
Parameters:
------------------------------------
```
GPT-4:
```
Parameters:
1. ?f - householdObject: the food to be heated
```
There is an inaccurate object type: a pan ?p can be viewed as a small receptacle
```
2. ?p - householdObject: the pan to be used for heating the food
3. ?s - furnitureAppliance: the stove burner on which the pan is placed

Preconditions:
(and
    (robot-at ?s)
```
Syntax error: the second parameter of `object-on` should
    be a furnitureAppliance, but a householdObject was given
```
    (object-on ?f ?p)
    (object-on ?p ?s)
```
Incorrect precondition: ?f needs not be pickupable
```
    (pickupable ?f)
    (is-small-receptacle ?p)
```
Incorrect precondition: a stove or burner ?s doesn't need to have a flat surface
```
    (flat-surface ?s)
    (robot-hand-empty)
```
Missing precondition: ?p has to be a pan
Missing precondition: ?s has to be a stove burner
```
)

Effects:
(and
    (not (pickupable ?f))
    (food-heated-pan ?f)
)
------------------------------------
```
Corrective feedback:
```
There are some errors in the PDDL model:
1. There is an inaccurate object type, a pan ?p can be viewed as a small receptacle
```

---

```
2. There is a syntax error: the second parameter of `object-on` should be a furnitureAppliance,
↪  but a householdObject was given. Please use the correct predicate or devise new one(s) if
↪  needed
3. Incorrect precondition: ?f needs not be pickupable
4. There is a missing precondition: ?p has to be a pan
5. There is a missing precondition: ?s has to be a stove burner
6. There is an incorrect precondition: a stove or burner ?s doesn't need to have a flat surface

Please revise the PDDL model (and the list of predicates if needed) to fix the above errors (and
↪  other potentially similar errors).

Parameters:
----------------------------------------
```

```
1. ?f - householdObject: the food to be heated
2. ?p - smallReceptacle: the pan to be used for heating the food
3. ?s - furnitureAppliance: the stove burner on which the pan is placed

Preconditions:
(and
    (robot-at ?s)
    (object-in ?f ?p)
    (object-on ?p ?s)
    (pan ?p)
    (stove-burner ?s)
    (robot-hand-empty)
)

Effects:
(and
    (not (pickupable ?f))
    (food-heated-pan ?f)
)

New Predicates:
1. (pan ?x - smallReceptacle): true if the small receptacle ?x is a pan
2. (stove-burner ?x - furnitureAppliance): true if the furniture or appliance ?x is a stove burner
3. (object-in ?x - householdObject ?y - smallReceptacle): true if the object ?x is in the small
↪  receptacle ?y
----------------------------------------
```

```
The following predicate(s) have the same name(s) as existing predicate(s):
1. (object-in ?x - householdObject ?y - smallReceptacle), true if the object ?x is in the small
↪  receptacle ?y; existing predicate with the same name: (object-in ?x - householdObject ?z -
↪  smallReceptacle), true if the object ?x is in the small receptacle ?z

You should reuse existing predicates whenever possible. If you are reusing existing predicate(s),
↪  you shouldn't list them under 'New Predicates'. If existing predicates are not enough and you
↪  are devising new predicate(s), please use names that are different from existing ones.

Please revise the PDDL model to fix this error.

Parameters:
----------------------------------------
```

```
1. ?f - householdObject: the food to be heated
2. ?p - smallReceptacle: the pan to be used for heating the food
3. ?s - furnitureAppliance: the stove burner on which the pan is placed

Preconditions:
(and
    (robot-at ?s)
    (object-in ?f ?p)
    (object-on ?p ?s)
    (pan ?p)
    (stove-burner ?s)
    (robot-hand-empty)
)

Effects:
(and
    (not (pickupable ?f))
    (food-heated-pan ?f)
)

New Predicates:
1. (pan ?x - smallReceptacle): true if the small receptacle ?x is a pan
2. (stove-burner ?x - furnitureAppliance): true if the furniture or appliance ?x is a stove burner
```

## A.7.2 Slice an object

```
GPT-4: Slice an object

User:
...
The original prompt for PDDL construction
...
Parameters:
------------------------------------
GPT-4:
Parameters:
1. ?x - householdObject: the object to slice
2. ?k - householdObject: the knife to use for slicing
3. ?y - furnitureAppliance: the furniture or appliance with a flat surface for manipulation
Incorrect object type, a cutting board should be viewed as
    a small receptacle as mentioned in the prompt
4. ?z - householdObject: the cutting board on which the object is placed

Preconditions:
(and
    (robot-at ?y)
    Incorrect precondition: ?x should be on the cutting board not directly on the furniture ?y
    (object-on ?x ?y)
    (object-on ?z ?y)
    (robot-holding ?k)
    (sliceable ?x)
    (cutting-board ?z)
    (knife ?k)
    (flat-surface ?y)
    (object-whole ?x)
)

Effects:
(and
    (not (object-whole ?x))
    (object-sliced ?x)
)
----------------------------------------
Corrective feedback:
There are some errors in the PDDL model:
1. There is an incorrect object type: a cutting board should be viewed as a small receptacle as
↪   mentioned in the prompt
2. There is an incorrect precondition: ?x should be on the cutting board not directly on the
↪   furniture ?y

Please revise the PDDL model (and the list of predicates if needed) to fix the above errors (and
↪   other potentially similar errors).

Parameters:
----------------------------------------
GPT-4:
1. ?x - householdObject: the object to slice
2. ?k - householdObject: the knife to use for slicing
3. ?y - furnitureAppliance: the furniture or appliance with a flat surface for manipulation
4. ?z - smallReceptacle: the cutting board on which the object is placed

Preconditions:
(and
    (robot-at ?y)
    (object-in ?x ?z)
    (object-on ?z ?y)
    (robot-holding ?k)
    (sliceable ?x)
    (cutting-board ?z)
    (knife ?k)
    (flat-surface ?y)
    (object-whole ?x)
)

Effects:
(and
    (not (object-whole ?x))
    (object-sliced ?x)
)

New Predicates:
No newly defined predicate
```

### A.7.3 GPT-3.5-Turbo trying to correct the action model of "slice an object"

---

**GPT-3.5-Turbo trying to correct the action model of "slice an object"**

User:
...
The original prompt for PDDL construction
...
Parameters:
----------------------------------
GPT-4:
Parameters:
1. ?x - householdObject: the object to slice
2. ?k - householdObject: the knife to use for slicing
3. ?y - furnitureAppliance: the furniture or appliance with a flat surface for manipulation
Incorrect object type, a cutting board should be viewed as
    a small receptacle as mentioned in the prompt
4. ?z - householdObject: the cutting board on which the object is placed

Preconditions:
(and
    (robot-at ?y)
    Incorrect precondition: ?x should be on the cutting board not directly on the furniture ?y
    (object-on ?x ?y)
    (object-on ?z ?y)
    (robot-holding ?k)
    (sliceable ?x)
    (cutting-board ?z)
    (knife ?k)
    (flat-surface ?y)
    (object-whole ?x)
)

Effects:
(and
    (not (object-whole ?x))
    (object-sliced ?x)
)
----------------------------------------
Corrective feedback:
There are some errors in the PDDL model:
1. There is an incorrect object type: a cutting board should be viewed as a small receptacle as
↪   mentioned in the prompt
2. There is an incorrect precondition: ?x should be on the cutting board not directly on the
↪   furniture ?y

Please revise the PDDL model (and the list of predicates if needed) to fix the above errors (and
↪   other potentially similar errors).

Parameters:
----------------------------------------
GPT-3.5-Turbo:
1. ?x - householdObject: the object to slice
2. ?k - householdObject: the knife to use for slicing
3. ?y - furnitureAppliance: the furniture or appliance with a flat surface for manipulation
4. ?z - smallReceptacle: the cutting board on which the object is placed

Preconditions:
(and
    (robot-at ?y)
    (object-on ?z ?y)
    (object-in ?x ?z)
    (robot-holding ?k)
    (sliceable ?x)
    Should use the predicate (cutting-board ?z)
    (is-small-receptacle ?z)
    (knife ?k)
    (flat-surface ?y)
    (object-whole ?x)
)

Effects:
(and
    (not (object-whole ?x))
    (object-sliced ?x)
)

## A.8 Logistics: Constructing PDDL Models

The AI agent here is a logistics planner that has to plan to transport packages within the locations in a city through a truck and between cities through an airplane. Within a city, the locations are directly linked, allowing trucks to travel between any two of these locations. Similarly, cities are directly connected to each other allowing airplanes to travel between any two cities. Each city is equipped with one truck and has a designated location that functions as an airport. There are five types of objects: package, truck, plane, location, and city. There are multiple cities and each city can have multiple locations. Also, there is no limit to how many packages a truck or plane can carry (so in theory a truck or plane can carry an infinite number of packages).

### A.8.1 Load a package into a truck

> **Action description**
>
> This action enables the agent to load a package into a truck. For example, load a package_1 into a
> ↪  truck_1.

> **GPT-4: Load a package into a truck**
>
> ```
> Parameters:
> 1. ?p - package: the package to load
> 2. ?t - truck: the truck to load the package into
> 3. ?l - location: the location where the package and truck are
>
> Preconditions:
> (and
>     (package-at ?p ?l)
>     (truck-at ?t ?l)
> )
>
> Effects:
> (and
>     (not (package-at ?p ?l))
>     (package-in-truck ?p ?t)
> )
> ```

### A.8.2 Unload a package from a truck

> **Action description**
>
> This action enables the agent to unload a package from a truck. For example, unload a package_1
> ↪  from a truck_1.

> **GPT-4: Unload a package from a truck**
>
> ```
> Parameters:
> 1. ?p - package: the package to unload
> 2. ?t - truck: the truck from which the package will be unloaded
> 3. ?l - location: the location where the package will be unloaded
>
> Preconditions:
> (and
>     (package-in-truck ?p ?t)
>     (truck-at ?t ?l)
> )
>
> Effects:
> (and
>     (not (package-in-truck ?p ?t))
>     (package-at ?p ?l)
> )
> ```

### A.8.3 Load a package into an airplane

**Action description**

This action enables the agent to load a package into an airplane. For example, load a package_1
↪ into an airplane_1.

**GPT-4: Load a package into an airplane**

```
Parameters:
1. ?p - package: the package to load into the airplane
2. ?a - plane: the airplane to load the package into
3. ?l - location: the location where the package and airplane are present

Preconditions:
(and
    (package-at ?p ?l)
    (plane-at ?a ?l)
    (not (package-in-plane ?p ?a))
)

Effects:
(and
    (not (package-at ?p ?l))
    (package-in-plane ?p ?a)
)
```

### A.8.4 Unload a package from an airplane

**Action description**

This action enables the agent to unload a package from an airplane. For example, unload a
↪ package_1 from an airplane_1.

**GPT-4: Unload a package from an airplane**

```
Parameters:
1. ?p - package: the package to unload
2. ?a - plane: the airplane from which the package will be unloaded
3. ?l - location: the location where the airplane is and the package will be unloaded

Preconditions:
(and
    (package-in-plane ?p ?a)
    (plane-at ?a ?l)
)

Effects:
(and
    (not (package-in-plane ?p ?a))
    (package-at ?p ?l)
)
```

### A.8.5 Drive a truck from one location to another in a city

**Action description**

This action enables the agent to drive a truck from one location to another in a city. For example,
↪ drive a truck_1 from location_1 to location_2 in city_1.

**GPT-4: Drive a truck from one location to another in a city**

```
Parameters:
1. ?t - truck: the truck to be driven
2. ?from - location: the starting location of the truck
3. ?to - location: the destination location of the truck
4. ?c - city: the city in which the truck is being driven

Preconditions:
```

```
(and
    (truck-at ?t ?from)
    (location-in-city ?from ?c)
    (location-in-city ?to ?c)
    (not (= ?from ?to))
)

Effects:
(and
    (not (truck-at ?t ?from))
    (truck-at ?t ?to)
    (move-packages ?t ?from ?to)
)
```

### A.8.6 Fly an airplane from one city to another

Action description

```
This action enables the agent to fly an airplane from one city's airport to another. The airports
↪  are locations in the city. For example, fly an airplane_1 from location_0 to location_1.
```

GPT-4: Fly an airplane from one city to another

```
Parameters:
1. ?a - plane: the airplane to fly
2. ?from - location: the departure location (airport) in the origin city
3. ?to - location: the destination location (airport) in the destination city
4. ?c1 - city: the origin city
5. ?c2 - city: the destination city

Preconditions:
(and
    (plane-at ?a ?from)
    (location-in-city ?from ?c1)
    (location-in-city ?to ?c2)
    (not (= ?c1 ?c2))
    Missing precondition: ?from should be an airport
    Missing precondition: ?to should be an airport
)

Effects:
(and
    (not (plane-at ?a ?from))
    (plane-at ?a ?to)
    (move-packages-plane ?a ?from ?to)
)
```

### A.8.7 Examples of PDDL action models constructed by GPT-3.5-Turbo

GPT-3.5-Turbo: Fly an airplane from one city to another

```
Parameters:
1. ?a - plane: the airplane to fly
2. ?c1 - city: the city to fly from
3. ?c2 - city: the city to fly to
4. ?l1 - location: the airport location in ?c1
5. ?l2 - location: the airport location in ?c2

Preconditions:
(and
    Incorrect predicate design
    (at ?a ?l1)
    (in-city ?l1 ?c1)
    (in-city ?l2 ?c2)
    (not (= ?c1 ?c2))
    Missing precondition: ?l1 should be an airport
    Missing precondition: ?l2 should be an airport
)

Effects:
(and
    Incorrect predicate design
```

```
        (not (at ?a ?l1))
        Incorrect predicate design
        (at ?a ?l2)
)
```

### GPT-3.5-Turbo: Unload a package from a truck

```
Parameters:
1. ?p - package: the package to unload
2. ?t - truck: the truck to unload from
Missing parameter: ?l, the location of the truck

Preconditions:
(and
    Syntax error and incorrect predicate design
    (at ?t - truck ?l - location)
    Syntax error and incorrect predicate design
    (at ?p - package ?t - truck)
)

Effects:
(and
    Syntax error
    (at ?p - package ?l - location)
    Syntax error
    (not (at ?p - package ?t - truck))
    Syntax error and incorrect effect
    (not (truck-empty ?t - truck))
)
```

### A.8.8  The initial set of predicates extracted by GPT-4

### The initial set of predicates extracted by GPT-4

```
1. (package-at ?p - package ?l - location): true if the package ?p is at the location ?l
2. (truck-at ?t - truck ?l - location): true if the truck ?t is at the location ?l
3. (package-in-truck ?p - package ?t - truck): true if the package ?p is in the truck ?t
4. (plane-at ?a - plane ?l - location): true if the airplane ?a is at the location ?l
5. (location-in-city ?l - location ?c - city): true if the location ?l is in the city ?c
6. (package-in-plane ?p - package ?a - plane): true if the package ?p is in the airplane ?a
7. (move-packages ?t - truck ?from - location ?to - location): true if all packages in the truck
↪  ?t are moved from location ?from to location ?to. This predicate is used to represent the
↪  effect of moving all packages in the truck when the truck is driven from one location to
↪  another.
8. (move-packages-plane ?a - plane ?from - location ?to - location): true if all packages in the
↪  airplane ?a are moved from location ?from to location ?to. This predicate is used to represent
↪  the effect of moving all packages in the airplane when the airplane is flown from one location
↪  to another
```

## A.9  Logistics: Correcting PDDL Models

### A.9.1  Fly an airplane from one city to another

### GPT-4: Fly an airplane from one city to another

```
User:
...
The original prompt for PDDL construction
...
Parameters:
------------------------------------
GPT-4:
Parameters:
1. ?a - plane: the airplane to fly
2. ?from - location: the departure location (airport) in the origin city
3. ?to - location: the destination location (airport) in the destination city
4. ?c1 - city: the origin city
5. ?c2 - city: the destination city

Preconditions:
(and
    (plane-at ?a ?from)
```

```
        (location-in-city ?from ?c1)
        (location-in-city ?to ?c2)
        (not (= ?c1 ?c2))
        Missing precondition: ?from should be an airport
        Missing precondition: ?to should be an airport
)

Effects:
(and
        (not (plane-at ?a ?from))
        (plane-at ?a ?to)
        (move-packages-plane ?a ?from ?to)
)
----------------------------------------
Corrective feedback:
There is an error in the PDDL model:
1. The current action model allows the plane to fly from any location to another location in a
↪ different city, but a plane can only land at airports

Please revise the PDDL model (and the list of predicates if needed) to fix the above errors (and
↪ other potentially similar errors).

Parameters:
----------------------------------------
GPT-4:
1. ?a - plane: the airplane to fly
2. ?from - location: the departure location (airport) in the origin city
3. ?to - location: the destination location (airport) in the destination city
4. ?c1 - city: the origin city
5. ?c2 - city: the destination city

Preconditions:
(and
        (plane-at ?a ?from)
        (location-in-city ?from ?c1)
        (location-in-city ?to ?c2)
        (not (= ?c1 ?c2))
        (airport ?from)
        (airport ?to)
)

Effects:
(and
        (not (plane-at ?a ?from))
        (plane-at ?a ?to)
        (move-packages-plane ?a ?from ?to)
)

New Predicates:
1. (airport ?l - location): true if the location ?l is an airport
```

## A.9.2 GPT-3.5-Turbo trying to correct the action model of "fly an airplane from one city to another"

GPT-3.5-Turbo trying to correct the action model of "fly an airplane from one city to another"

```
User:
...
The original prompt for PDDL construction
...
Parameters:
------------------------------------
GPT-4:
Parameters:
1. ?a - plane: the airplane to fly
2. ?from - location: the departure location (airport) in the origin city
3. ?to - location: the destination location (airport) in the destination city
4. ?c1 - city: the origin city
5. ?c2 - city: the destination city

Preconditions:
(and
        (plane-at ?a ?from)
        (location-in-city ?from ?c1)
        (location-in-city ?to ?c2)
        (not (= ?c1 ?c2))
```

```
        Missing precondition: ?from should be an airport
        Missing precondition: ?to should be an airport
)

Effects:
(and
    (not (plane-at ?a ?from))
    (plane-at ?a ?to)
    (move-packages-plane ?a ?from ?to)
)
----------------------------------------
Corrective feedback:
There is an error in the PDDL model:
1. The current action model allows the plane to fly from any location to another location in a
↪    different city, but a plane can only land at airports

Please revise the PDDL model (and the list of predicates if needed) to fix the above errors (and
↪    other potentially similar errors).

Parameters:
----------------------------------------
GPT-3.5-Turbo:
1. ?a - plane: the airplane to fly
2. ?from - location: the departure location (airport) in the origin city
3. ?to - location: the destination location (airport) in the destination city
4. ?c1 - city: the origin city
5. ?c2 - city: the destination city

Preconditions:
(and
    (plane-at ?a ?from)
    (location-in-city ?from ?c1)
    (location-in-city ?to ?c2)
    (not (= ?c1 ?c2))
    (airport ?to)
    Missing precondition: ?from should be an airport
)

Effects:
(and
    (not (plane-at ?a ?from))
    (plane-at ?a ?to)
    (move-packages-plane ?a ?from ?to)
)

New Predicates:
1. (airport ?l - location): true if the location ?l is an airport.
```

## A.10 Tyreworld: Constructing PDDL Models

The AI agent here is a robot that has to replace a flat tyre with a spare one. This involves fetching the tools (i.e., wrench, jack, pump) from the boot, undoing the nuts on the flat tyre, jacking up the (appropriate) hub(s), removing the tyre, doing up the spare one, etc. There are three major object types: 'small_object', 'container' and 'hub'. The object type 'small_object' covers tools, wheels and nuts. The 'small_object' object type has three subtypes: 'tool', 'wheel' and 'nut'. The subtype 'tool' covers tools like wrenches, jacks, pumps etc. The subtype 'wheel' covers various wheels. The subtype 'nut' covers various nuts. The object type 'container' covers storage spaces like the boot in a car. The object type 'hub' covers the hubs in the wheels of the car. Note that there is no restriction on how many objects the AI agent (i.e., the robot) can carry. Also note that each hub has only one nut.

### A.10.1 Open a container

**Action description**

```
This action enables the robot to open a container. For example, the robot can open a boot.
```

**GPT-4: Open a container**

```
Parameters:
1. ?c - container: the container to open
```

```
Preconditions:
(and
    (container-closed ?c)
)

Effects:
(and
    (not (container-closed ?c))
    (container-open ?c)
)
```

### A.10.2 Close a container

Action description

```
This action enables the robot to close a container. For example, the robot can close a boot.
```

GPT-4: Close a container

```
Parameters:
1. ?c - container: the container to close

Preconditions:
(and
    (container-open ?c)
)

Effects:
(and
    (not (container-open ?c))
    (container-closed ?c)
)
```

### A.10.3 Fetch an object from a container

Action description

```
This action enables the robot to take an object from a container. For example, the robot can fetch
↪  jack from the boot or the robot can fetch wheel_1 from the boot.
```

GPT-4: Fetch an object from a container

```
Parameters:
1. ?x - small_object: the object to fetch from the container
2. ?c - container: the container from which the object is fetched

Preconditions:
(and
    (container-open ?c)
    (object-in-container ?x ?c)
)

Effects:
(and
    (not (object-in-container ?x ?c))
    (robot-holding ?x)
)
```

### A.10.4 Put an object into a container

Action description

```
This action enables the robot to put an object into a container. For example, the robot can put
↪  away jack into the boot or the robot can put away wheel_1 into the boot.
```

```
Parameters:
1. ?x - small_object: the object to put into the container
2. ?c - container: the container to put the object into

Preconditions:
(and
    (robot-holding ?x)
    (container-open ?c)
)

Effects:
(and
    (not (robot-holding ?x))
    (object-in-container ?x ?c)
)
```

### A.10.5 Loosen a nut in a hub

Action description

This action enables the robot to loosen a nut in a hub. For example, the robot can loosen nut_1 in
↪ hub_1. To safely perform this action, the hub needs to be on the ground (i.e., not jacked up).

GPT-4: Loosen a nut in a hub

```
Parameters:
1. ?n - nut: the nut to be loosened
2. ?h - hub: the hub where the nut is located
3. ?w - tool: the wrench to be used for loosening the nut

Preconditions:
(and
    (is-nut ?n)
    (is-hub ?h)
    (is-wrench ?w)
    (nut-in-hub ?n ?h)
    (hub-on-ground ?h)
    (robot-holding ?w)
)

Effects:
(and
    (nut-loose ?n)
)
```

### A.10.6 Tighten a nut in a hub

Action description

This action enables the robot to tighten a nut in a hub. For example, the robot can tighten nut_1
↪ in hub_1. To safely perform this action, the hub needs to be on the ground (i.e., not jacked
↪ up).

GPT-4: Tighten a nut in a hub

```
Parameters:
1. ?n - nut: the nut to tighten
2. ?h - hub: the hub where the nut is to be tightened
3. ?w - tool: the wrench to use for tightening the nut

Preconditions:
(and
    (is-nut ?n)
    (is-hub ?h)
    (is-wrench ?w)
    (robot-holding ?w)
    (nut-in-hub ?n ?h)
    (hub-on-ground ?h)
    (nut-loose ?n)
```

```
)

Effects:
(and
    (not (nut-loose ?n))
)
```

### A.10.7 Jack up a hub

**Action description**

This action enables the robot to jack up a hub. For example, the robot can jack up hub_1.

**GPT-4: Jack up a hub**

```
Parameters:
1. ?h - hub: the hub to jack up
2. ?j - small_object: the jack to use for jacking up the hub

Preconditions:
(and
    (is-hub ?h)
    (hub-on-ground ?h)
    (is-jack ?j)
    (robot-holding ?j)
)

Effects:
(and
    (not (hub-on-ground ?h))
    (hub-jacked-up ?h)
)
```

### A.10.8 Jack down a hub

**Action description**

This action enables the robot to jack down a hub. For example, the robot can jack down hub_1.

**GPT-4: Jack down a hub**

```
Parameters:
1. ?h - hub: the hub to jack down
2. ?j - small_object: the jack used to jack down the hub

Preconditions:
(and
    (is-hub ?h)
    (is-jack ?j)
    (robot-holding ?j)
    (hub-jacked-up ?h)
)

Effects:
(and
    (not (hub-jacked-up ?h))
    (hub-on-ground ?h)
)
```

### A.10.9 Unfasten a hub

**Action description**

This action enables the robot to unfasten a hub and remove the nut on it. For example, the robot
↪  can unfasten hub_1 that has nut_1. At the end of this action, the hub should be unfastened,
↪  and the nut should be detached from the hub and held by the robot. To safely perform this
↪  action, the hub needs to be jacked up.

```

## GPT-4: Unfasten a hub

```
Parameters:
1. ?h - hub: the hub to unfasten
2. ?n - nut: the nut to detach from the hub

Preconditions:
(and
    (is-hub ?h)
    (is-nut ?n)
    (nut-in-hub ?n ?h)
    (nut-loose ?n)
    (hub-jacked-up ?h)
    Incorrect precondition: the robot should not hold the nut if the nut is in the hub
    (robot-holding ?n)
)

Effects:
(and
    (not (nut-in-hub ?n ?h))
    (hub-unfastened ?h)
    Missing effect: the robot should hold the nut at the end
)
```

### A.10.10  Fasten a hub

#### Action description

```
This action enables the robot to fasten a hub by installing a nut on it. For example, the robot
↪  can fasten hub_1 that has nut_1. Note that the robot needs to install a nut into the hub, but
↪  the robot is not required to fasten the nut in this operation. To safely perform this action,
↪  the hub needs to be jacked up.
```

#### GPT-4: Fasten a hub

```
Parameters:
1. ?h - hub: the hub to fasten
2. ?n - nut: the nut to install in the hub

Preconditions:
(and
    (is-hub ?h)
    (is-nut ?n)
    (robot-holding ?n)
    (hub-jacked-up ?h)
    (hub-unfastened ?h)
    Unnecessary precondition
    (not (nut-in-hub ?n ?h))
)

Effects:
(and
    (not (robot-holding ?n))
    (nut-in-hub ?n ?h)
    (not (hub-unfastened ?h))
    Missing effect: the nut should be loose
)
```

### A.10.11  Remove wheel from hub

#### Action description

```
This action enables the robot to remove a wheel from a hub. For example, the robot can remove
↪  wheel_1 from hub_1. Note that the hub needs to be unfastened in order to remove the wheel. To
↪  safely perform this action, the hub needs to be jacked up.
```

#### GPT-4: Remove wheel from hub

```
Parameters:
1. ?w - wheel: the wheel to remove
2. ?h - hub: the hub from which the wheel is to be removed
```

```
Preconditions:
(and
    (is-wheel ?w)
    (is-hub ?h)
    (wheel-on-hub ?w ?h)
    (hub-jacked-up ?h)
    (hub-unfastened ?h)
)

Effects:
(and
    (not (wheel-on-hub ?w ?h))
    (robot-holding ?w)
)
```

### A.10.12 Put wheel on hub

**Action description**

```
This action enables the robot to put a wheel on an unfastened hub. For example, the robot can put
↪  wheel_1 on hub_1. Note that the hub needs to be unfastened in order to put on the wheel. To
↪  safely perform this action, the hub needs to be jacked up.
```

**GPT-4: Put wheel on hub**

```
Parameters:
1. ?w - wheel: the wheel to put on the hub
2. ?h - hub: the hub to put the wheel on

Preconditions:
(and
    (robot-holding ?w)
    (is-wheel ?w)
    (is-hub ?h)
    (hub-jacked-up ?h)
    (hub-unfastened ?h)
    (not (wheel-on-hub ?w ?h))
)

Effects:
(and
    (not (robot-holding ?w))
    (wheel-on-hub ?w ?h)
)
```

### A.10.13 Inflate wheel

**Action description**

```
This action enables the robot to inflate a wheel. For example, the robot can inflate wheel_1. Note
↪  that you need to make sure the wheel is intact.
```

**GPT-4: Inflate wheel**

```
Parameters:
1. ?w - wheel: the wheel to inflate

Preconditions:
(and
    (is-wheel ?w)
    (robot-holding ?w)
    (wheel-deflated ?w)
    (wheel-intact ?w)
    Missing precondition: the robot should have a pump
)

Effects:
(and
    (not (wheel-deflated ?w))
    (wheel-inflated ?w)
)
```

### A.10.14 Examples of PDDL action models constructed by GPT-3.5-Turbo

---

**GPT-3.5-Turbo: Open a containe**

```
Parameters:
1. ?x - container: the container to open

Preconditions:
(and
    (not (container-open ?x))
    Incorrect precondition and unknown parameter
    (robot-holding ?y)
    Incorrect precondition and syntax error
    (tool-type ?y "key")
    Incorrect precondition and syntax error
    (not (hub-jacked-up ?z))
)

Effects:
(and
    (container-open ?x)
    Incorrect effect
    (not (robot-holding ?y))
)
```

---

**GPT-3.5-Turbo: Loosen a nut in a hub**

```
Parameters:
1. ?x - nut: the nut to loosen
2. ?y - hub: the hub containing the nut
Missing parameter: ?z, the wrench

Preconditions:
(and
    (not (hub-jacked-up ?y))
    (robot-holding ?z)
    Syntax error
    (tool-type ?z wrench)
    (nut-in-hub ?x ?y)
    (not (nut-loosened ?x))
    Irrelevant precondition
    (not (wheel-detached ?w))
    Irrelevant precondition
    (wheel-attached ?w ?y)
    Irrelevant precondition
    (wheel-intact ?w)
    Irrelevant precondition
    (not (wheel-deflated ?w))
    Irrelevant precondition
    (object-in-container ?z ?c)
    Irrelevant precondition
    (container-open ?c)
)

Effects:
(and
    (nut-loosened ?x)
)
```

---

### A.10.15 The initial set of predicates extracted by GPT-4

**The initial set of predicates extracted by GPT-4**

```
1. (container-closed ?c - container): true if the container ?c is closed
2. (container-open ?c - container): true if the container ?c is open
3. (object-in-container ?x - small_object ?c - container): true if the small_object ?x is inside
↪   the container ?c
4. (robot-holding ?x - small_object): true if the robot is holding the small_object ?x
5. (is-wrench ?x - small_object): true if the small_object ?x is a wrench
6. (is-nut ?x - small_object): true if the small_object ?x is a nut
7. (is-hub ?x - hub): true if the object ?x is a hub
8. (nut-in-hub ?n - nut ?h - hub): true if the nut ?n is in the hub ?h
9. (hub-on-ground ?h - hub): true if the hub ?h is on the ground (not jacked-up)
10. (nut-loose ?n - nut): true if the nut ?n is loosened
```

```
11. (is-jack ?x - small_object): true if the small_object ?x is a jack
12. (hub-jacked-up ?h - hub): true if the hub ?h is jacked up (not on the ground)
13. (hub-unfastened ?h - hub): true if the hub ?h is unfastened
14. (is-wheel ?x - small_object): true if the small_object ?x is a wheel
15. (wheel-on-hub ?w - wheel ?h - hub): true if the wheel ?w is on the hub ?h
16. (is-pump ?x - small_object): true if the small_object ?x is a pump
17. (wheel-intact ?w - wheel): true if the wheel ?w is intact
18. (wheel-deflated ?w - wheel): true if the wheel ?w is deflated
19. (wheel-inflated ?w - wheel): true if the wheel ?w is inflated
```

## A.11  Tyreworld: Correcting PDDL Models

### A.11.1  Unfasten a hub

---

**GPT-4: Unfasten a hub**

User:
...
The original prompt for PDDL construction
...
Parameters:
------------------------------------
GPT-4:
Parameters:
1. ?h - hub: the hub to unfasten
2. ?n - nut: the nut to detach from the hub

Preconditions:
(and
    (is-hub ?h)
    (is-nut ?n)
    (nut-in-hub ?n ?h)
    (nut-loose ?n)
    (hub-jacked-up ?h)
    Incorrect precondition: the robot should not hold the nut if the nut is in the hub
    (robot-holding ?n)
)

Effects:
(and
    (not (nut-in-hub ?n ?h))
    (hub-unfastened ?h)
    Missing effect: the robot should hold the nut at the end
)
---------------------------------------
Corrective feedback:
There are some errors in the PDDL model:
1. There is an incorrect precondition: the robot is not holding the nut because the nut is in the
↪   hub
2. There is a missing effect: the robot should hold the nut at the end

Please revise the PDDL model (and the list of predicates if needed) to fix the above errors (and
↪   other potentially similar errors).

Parameters:
---------------------------------------
GPT-4:
1. ?h - hub: the hub to unfasten
2. ?n - nut: the nut to detach from the hub

Preconditions:
(and
    (is-hub ?h)
    (is-nut ?n)
    (nut-in-hub ?n ?h)
    (nut-loose ?n)
    (hub-jacked-up ?h)
)

Effects:
(and
    (not (nut-in-hub ?n ?h))
    (hub-unfastened ?h)
    (robot-holding ?n)
)
```

---

## A.11.2 Inflate wheel

---

**GPT-4: Inflate wheel**

```
...
The original prompt for PDDL construction
...
Parameters:
-------------------------------------
```
GPT-4:
```
Parameters:
1. ?w - wheel: the wheel to inflate

Preconditions:
(and
    (is-wheel ?w)
    (robot-holding ?w)
    (wheel-deflated ?w)
    (wheel-intact ?w)
    Missing precondition: the robot should have a pump
)

Effects:
(and
    (not (wheel-deflated ?w))
    (wheel-inflated ?w)
)
-----------------------------------------
```
Corrective feedback:
```
There is an error in the PDDL model:
1. You allow the robot to inflate a wheel without having a pump, this is incorrect.

Please revise the PDDL model (and the list of predicates if needed) to fix the above errors (and
↪   other potentially similar errors).

Parameters:
-----------------------------------------
```
GPT-4:
```
1. ?w - wheel: the wheel to inflate
2. ?p - pump: the pump to inflate the wheel

Preconditions:
(and
    (is-wheel ?w)
    (is-pump ?p)
    (robot-holding ?w)
    (robot-holding ?p)
    (wheel-deflated ?w)
    (wheel-intact ?w)
)

Effects:
(and
    (not (wheel-deflated ?w))
    (wheel-inflated ?w)
)

New Predicates:
No newly defined predicate
-----------------------------------------
```
Automated corrective feedback:
```
There is an error in the PDDL model:
1. There is an unknown object type `pump` for the parameter ?p

Please revise the PDDL model (and the list of predicates if needed) to fix the above errors (and
↪   other potentially similar errors).

Parameters:
-----------------------------------------
```
GPT-4:
```
1. ?w - wheel: the wheel to inflate
2. ?p - small_object: the pump to inflate the wheel

Preconditions:
(and
    (is-wheel ?w)
    (is-pump ?p)
```

```
    (robot-holding ?w)
    (robot-holding ?p)
    (wheel-deflated ?w)
    (wheel-intact ?w)
)

Effects:
(and
    (not (wheel-deflated ?w))
    (wheel-inflated ?w)
)
```

## A.12   LLM planners back-prompted by VAL using LLM-acquired PDDL model

## A.12.1   Examples of prompts for LLM planners

An example prompt for the Household domain

```
You are in a household to complete a certain task for the owners. You are a household robot that
↪  can navigate to various large and normally immovable furniture pieces or appliances in the
↪  house to carry out household tasks. Note that you have only one gripper, so (a) it can only
↪  hold one object, (b) it shouldn't hold any other irrelevant objects in its gripper while
↪  performing some manipulation tasks (e.g., opening a drawer or closing a window), (c)
↪  operations on small household items should be carried out on furniture with a flat surface to
↪  get enough space for manipulation.

The following actions are available to you, and you should strictly follow the output format
↪  (demonstrated in the "example output") of each action:

1. navigate from a furniture piece or appliance ?x to another ?y (example output: go from
↪  dining_table_1 to fridge_1).
Parameters: (a) ?x: the furniture or appliance the robot is currently at; (b) ?y: the furniture or
↪  appliance the robot wants to navigate to.
Preconditions: (a) the robot is at the furniture or appliance ?x; (b) the furniture or appliance
↪  ?x is not equal to ?y.
Effects: (a) the robot is not at the furniture or appliance ?x; (b) the robot is at the furniture
↪  or appliance ?y.

2. pick up an object ?x in/on a furniture piece or appliance ?y (example output: pick up apple_1
↪  in/on side_table_1).
Parameters: (a) ?x: the object to pick up; (b) ?y: the furniture or appliance where the object is
↪  located.
Preconditions: (a) the robot is at the furniture or appliance ?y; (b) the object ?x is on or in
↪  the furniture or appliance ?y; (c) the object ?x can be picked up by the robot; (d) either the
↪  object ?x cannot be stacked on top of other household items, or if it can be stacked, it is
↪  not stacked on top of and not under any another household object; (e) the robot's gripper is
↪  empty; (f) either the furniture or appliance ?y cannot be opened, or it is already opened.
Effects: (a) the object ?x is not on or in the furniture or appliance ?y; (b) the robot is holding
↪  the object ?x; (c) the robot's gripper is not empty.
...

Here are two examples.

Example 1
The house contains the following large furniture pieces and appliances: drawer_1, fridge_1,
↪  dining_table_1, countertop_1, stove_burner_1, cabinet_1, microwave_1, sink_basin_1,
↪  dish_washer_1, light_switch_1, garbage_can_1.

The current states of the household objects are as follows:
1. drawer_1 is closed
2. fridge_1 is closed
...
17. robot is not holding anything

Your task is to: put an orange in a fridge.
Your plan is:

Step 1:
> think: First, I need to open the fridge.
> action: go from dining_table_1 to fridge_1
Step 2:
> action: open furniture/appliance fridge_1
Step 3:
> think: Now I need to find an orange. In the current state, orange_1 is in drawer_1 and drawer_1
↪  is closed. I need to get the orange and then put it in the fridge.
> action: go from fridge_1 to drawer_1
Step 4:
```

```
> action: open furniture/appliance drawer_1
Step 5:
> action: pick up orange_1 in/on drawer_1
Step 6:
> action: go from drawer_1 to fridge_1
Step 7:
> action: put orange_1 in/on fridge_1
Step 8:
> action: close furniture/appliance fridge_1

Example 2
...

Here is the task.
The house contains the following large furniture pieces and appliances: drawer_1, drawer_2, ...,
↪   dish_washer_1, garbage_can_1. Note that cutting boards are viewed as small receptacles, and
↪   blenders are small appliances (and thus small receptacles).

The current states of the household objects are as follows:
1. drawer_1 is opened
2. drawer_2 is opened
...
71. robot is holding mug_1

Your task is to: put apple_2 on side_table_2.
Your plan is:
```

## An example prompt for the Logistics domain

```
You are a logistics planner that has to plan to transport packages within the locations in a city
↪   through a truck and between cities through an airplane. Within a city, the locations are
↪   directly linked, allowing trucks to travel between any two of these locations. Similarly,
↪   cities are directly connected to each other allowing airplanes to travel between any two
↪   cities. Each city is equipped with one truck and has a designated location that functions as
↪   an airport. There are five types of objects: package, truck, plane, location, and city. There
↪   are multiple cities and each city can have multiple locations. Also, there is no limit to how
↪   many packages a truck or plane can carry (so in theory a truck or plane can carry an infinite
↪   number of packages).

The following actions are available to you, and you should strictly follow the output format
↪   (demonstrated in the "example output") of each action:

1. load a package ?p into a truck ?t at location ?l (example output: load package_1 into a truck
↪   truck_1 at location_0).
Parameters: (a) ?p: the package to load; (b) ?t: the truck to load the package into; (c) ?l: the
↪   location where the package and truck are.
Preconditions: (a) the package ?p is at the location ?l; (b) the truck ?t is at the location ?l.
Effects: (a) the package ?p is not at the location ?l; (b) the package ?p is in the truck ?t.

2. unload a package ?p from a truck ?t at location ?l (example output: unload package_1 from a
↪   truck truck_1 at location_1).
Parameters: (a) ?p: the package to unload; (b) ?t: the truck from which the package will be
↪   unloaded; (c) ?l: the location where the package will be unloaded.
Preconditions: (a) the package ?p is in the truck ?t; (b) the truck ?t is at the location ?l.
Effects: (a) the package ?p is not in the truck ?t; (b) the package ?p is at the location ?l.
...

Here are two examples.

Example 1
The environment contains the following objects: plane_0 (type: plane), city_0 (type: city),
↪   truck_0 (type: truck), location_0 (type: location), location_1 (type: location), package_0
↪   (type: package).

The current states of the objects are as follows:
1. location_0 is in city city_0
2. location_1 is in city city_0
3. truck_0 is at location location_0
4. plane_0 is at location location_0
5. package_0 is at location location_1
6. location_0 is an airport

Your task is to: Transport package package_0 to location location_0
Your plan is:
```

```
Step 1:
> think: First, I need to drive the truck to the location of the package and load the package.
> action: drive truck_0 from location_0 to location_1 in city_0
Step 2:
> action: load package_0 into a truck truck_0 at location_1
Step 3:
> think: Now I need to drive the truck to the destination and unload the package.
> action: drive truck_0 from location_1 to location_0 in city_0
Step 4:
> action: unload package_0 from a truck truck_0 at location_0

Example 2
...

Here is the task.
The environment contains the following objects: plane_0 (type: plane), city_0 (type: city), city_1
↪  (type: city), truck_0 (type: truck), truck_1 (type: truck), location_0 (type: location),
↪  location_1 (type: location), location_2 (type: location), package_0 (type: package)

The current states of the objects are as follows:
1. plane_0 is at location location_1
2. truck_0 is at location location_2
3. truck_1 is at location location_0
4. location_0 is in city city_1
5. location_0 is an airport
6. location_1 is in city city_0
7. location_1 is an airport
8. location_2 is in city city_0
9. package_0 is at location location_0

Your task is to: Transport package package_0 to location location_2.
Your plan is:
```

### A.12.2   Translation to admissible actions

We extract the plan from the LLM planner as a completion to the prompt. Although the prompt explicitly states that the planner should strictly adhere to the demonstrated output format of each action, there are still cases where GPT-4 fails to follow these formats. As a result, extra post-processing is needed to translate the generated actions into admissible actions. To do this, we utilize another LLM (i.e., GPT-4) to perform the translation. In cases where certain actions have missing parameters, we present an error message to the LLM planner and ask it to regenerate the plan. The error message is structured as follows: "There is an invalid output at step x. Please strictly follow the output format provided in the example output of each action.  Your revised plan:".

We use a fixed prompt template for action translation in all the domains:

Template of the prompt for action translation

```
Your task is to translate free-form action description to its corresponding structural format
↪  based on its semantic meaning. The structural output format is given in the example output of
↪  each candidate action. You should strictly follow the structural output format. When there is
↪  missing information in the free-form action description, you should output "Not able to
↪  translate due to missing information".

Here are 4 examples.
Example 1:
Action description: put bowl_1 with sliced orange_1 in/on dining_table_1
Candidate actions:
1. put an object on or in a furniture piece or an appliance, example output: "put apple_1 in/on
↪  side_table_1"
2. puts an object onto or into a small receptacle, example output: "put apple_1 in/on plate_1 at
↪  countertop_1"
Translated action: put bowl_1 in/on side_table_1

Example 2:
Action description: put sliced orange_1 in/on bowl_1 at drawer_1
Candidate actions:
1. put an object on or in a furniture piece or an appliance, example output: "put apple_1 in/on
↪  side_table_1"
2. puts an object onto or into a small receptacle, example output: "put apple_1 in/on plate_1 at
↪  countertop_1"
```

```
Translated action: put orange_1 in/on bowl_1 at drawer_1

Example 3:
Action description: pick up bowl_1 with sliced orange_1 in/on countertop_1
Candidate actions:
1. pick up an object on or in a furniture piece or an appliance, example output: "pick up apple_1
↪  in/on side_table_1"
2. pick up an object on or in a small receptacle, example output: "pick up apple_1 in/on plate_1
↪  at countertop_1"
Translated action: pick up bowl_1 in/on countertop_1

Example 4:
Action description: put mashed apple_1 in/on plate_1
Candidate actions:
1. put an object on or in a furniture piece or an appliance, example output: "put apple_1 in/on
↪  side_table_1"
2. puts an object onto or into a small receptacle, example output: "put apple_1 in/on plate_1 at
↪  countertop_1"
Translated action: Not able to translate due to missing information

Here is the task.
```

### A.12.3  Back-prompting LLM planners with validation feedback by VAL

We consider three types of validation feedback that can be obtained from VAL [18]. One is unsatisfied precondition. VAL highlights unsatisfied precondition(s) in PDDL, which can be translated into natural language and provided as corrective feedback to the LLM planner. The feedback message is structured as follows: "The action at step x is not executable due to unmet precondition(s). Here are the unsatisfied precondition(s):". The second type of validation feedback is related to unmet goal condition(s), and the third type is related to the usage of invalid or illegal parameter(s) in some action(s).

For translating PDDL into natural language, we use a fixed prompt template in all the domains:

---

**Template of the prompt for translating PDDL into natural language**

```
Your task is to translate planning domain definition language (PDDL) to natural language.

Here is an example:

Predicates:
1. (robot-holding ?x - block): true if the robot arm is holding the block ?x
2. (block-clear ?x - block): true if the block ?x is not under any another block
3. (robot-hand-empty): true if the robot arm is not holding any block
4. (block-on-table ?x - block): true if the block ?x is placed on the table

Contextual information: N/A

PDDL:
```
(and
    (not (robot-holding x1))
    (block-clear x1)
    (robot-hand-empty)
    (block-on-table x1)
)
```

Translated PDDL:
(and
    The robot arm is not holding the block x1;
    The block x1 is not under any another block;
    The robot arm is not holding any block;
    The block x1 is placed on the table;
)

Here is the task:
```

### A.12.4 Examples of validation feedback and instructions with ordering constraints

---

**Make heated mashed potato_1 & Remember to mash the potato at the end after heating it**

```
User:
...
The original prompt for the LLM planner
...
The current states of the household objects are as follows:
...
70. robot at dining_table_1
71. robot is holding mug_1

Your task is to: make heated and mashed potato_1, remember to mash the potato at the end after
↪  heating it.
Your plan is:
------------------------------------
GPT-4:
Step 1:
> think: First, I need to find a cutting board and put it on a flat surface (countertop_1 or
↪  countertop_2). In the current state, cutting_board_1 is in/on countertop_2.
> action: go from dining_table_1 to countertop_2
Step 2:
> think: Now I need to find potato_1. In the current state, potato_1 is in fridge_1 and fridge_1
↪  is closed. I need to get the potato and then put it on the cutting board.
> action: go from countertop_2 to fridge_1
Step 3:
> action: open furniture/appliance fridge_1
Step 4:
> action: pick up potato_1 in/on fridge_1
Step 5:
> action: go from fridge_1 to countertop_2
Step 6:
> action: put potato_1 in/on cutting_board_1 at countertop_2
Step 7:
> think: Now I need to find a knife. In the current state, knife_1 is in cabinet_3 and cabinet_3
↪  is closed. I need to get the knife and then slice the potato.
> action: go from countertop_2 to cabinet_3
Step 8:
> action: open furniture/appliance cabinet_3
Step 9:
> action: pick up knife_1 in/on cabinet_3
Step 10:
> action: go from cabinet_3 to countertop_2
Step 11:
> action: slice potato_1 with knife_1 and cutting_board_1 at countertop_2
Step 12:
> think: Now I need to heat the sliced potato. In the current state, microwave_1 is closed. I need
↪  to put the sliced potato in a microwave-safe container and then heat it in the microwave.
> action: go from countertop_2 to cabinet_4
Step 13:
> action: open furniture/appliance cabinet_4
Step 14:
> action: pick up bowl_2 in/on dish_washer_1
Step 15:
> action: go from cabinet_4 to countertop_2
Step 16:
> action: transfer potato_1 from cutting_board_1 to bowl_2 at countertop_2
Step 17:
> action: go from countertop_2 to microwave_1
Step 18:
> action: open furniture/appliance microwave_1
Step 19:
> action: put bowl_2 in/on microwave_1
Step 20:
> action: close furniture/appliance microwave_1
Step 21:
> action: heat potato_1 in bowl_2 with microwave_1
Step 22:
> action: open furniture/appliance microwave_1
Step 23:
> action: pick up bowl_2 in/on microwave_1
Step 24:
> action: go from microwave_1 to countertop_1
Step 25:
> think: Now I need to find a blender to mash the heated potato. In the current state, blender_1
↪  is in/on countertop_1 and blender_1 is turned off. I need to put the heated potato in the
↪  blender and then mash it.
```

```
> action: transfer potato_1 from bowl_2 to blender_1 at countertop_1
Step 26:
> action: mash potato_1 with blender_1 at countertop_1
Step 27:
> think: Task complete. The heated mashed potato_1 is now in the blender.
-----------------------------------
Validation feedback:
The action at step 3 is not executable due to unmet precondition(s). Here are the unsatisfied
↪   precondition(s):
The robot's gripper is not holding any object.

Your revised plan is:
-----------------------------------
GPT-4:
... The revised plan ...
```

## Place fork_1 spoon_1 and knife_1 on dining_table_1 & Please take the fork first and the knife last

```
User:
...
The original prompt for the LLM planner
...
The current states of the household objects are as follows:
...
24. lunch_box_2 in/on fridge_1
25. lunch_box_2 is not opened
...
59. spoon_1 in/on lunch_box_2
...

Your task is to: place fork_1, spoon_1, and knife_1 on dining_table_1, please take the fork first
↪   and the knife last.
Your plan is:
-----------------------------------
GPT-4:
Step 1:
> think: First, I need to put down the mug_1 I am currently holding.
> action: go from dining_table_1 to side_table_1
Step 2:
> action: put mug_1 in/on side_table_1
Step 3:
> think: Now, I need to pick up fork_1 from dish_washer_1.
> action: go from side_table_1 to dish_washer_1
Step 4:
> action: open furniture/appliance dish_washer_1
Step 5:
> action: pick up fork_1 in/on dish_washer_1
Step 6:
> think: I have the fork_1, now I need to place it on dining_table_1.
> action: go from dish_washer_1 to dining_table_1
Step 7:
> action: put fork_1 in/on dining_table_1
Step 8:
> think: Next, I need to pick up spoon_1 from lunch_box_2 in fridge_1.
> action: go from dining_table_1 to fridge_1
Step 9:
> action: open furniture/appliance fridge_1
Step 10:
> action: pick up spoon_1 in/on lunch_box_2 at fridge_1
Step 11:
> think: I have the spoon_1, now I need to place it on dining_table_1.
> action: go from fridge_1 to dining_table_1
Step 12:
> action: put spoon_1 in/on dining_table_1
Step 13:
> think: Finally, I need to pick up knife_1 from cabinet_1.
> action: go from dining_table_1 to cabinet_1
Step 14:
> action: pick up knife_1 in/on cabinet_1
Step 15:
> think: I have the knife_1, now I need to place it on dining_table_1.
> action: go from cabinet_1 to dining_table_1
Step 16:
> action: put knife_1 in/on dining_table_1
Step 17:
```

```
> think: All utensils are placed on the dining table. Now, I need to close the opened appliances.
> action: go from dining_table_1 to fridge_1
Step 18:
> action: close furniture/appliance fridge_1
Step 19:
> action: go from fridge_1 to dish_washer_1
Step 20:
> action: close furniture/appliance dish_washer_1
------------------------------------
Validation feedback:
The action at step 10 is not executable due to unmet precondition(s). Here are the unsatisfied
↪  precondition(s) described in Pseudo-PDDL format:
(and
    The furniture or appliance fridge_1 has a flat surface for manipulation;
    (or
        The small receptacle lunch_box_2 cannot be opened;
        The small receptacle lunch_box_2 is open;
    )
)

Your revised plan is:
------------------------------------
GPT-4:
... The revised plan ...
```

## A.13   Translating user instructions into PDDL goal specifications

### Template of the prompt for translating user instructions into PDDL goal specifications

```
Your task is to translate natural language instructions to planning goals in planning domain
↪  definition language (PDDL). You should only use the given predicates to express the goal
↪  specification.

Here is an example from the classical BlocksWorld domain for demonstrating the output format.

Domain information: BlocksWorld is a planning domain in artificial intelligence. The AI agent here
↪  is a mechanical robot arm that can pick and place the blocks. Only one block may be moved at a
↪  time: it may either be placed on the table or placed atop another block. Because of this, any
↪  blocks that are, at a given time, under another block cannot be moved. There is only one type
↪  of object in this domain, and that is the block.

The domain has the following objects: block_1 (type: block), block_2 (type: block), block_3 (type:
↪  block), block_4 (type: block), block_5 (type: block).

Predicates:
1. (robot-holding ?x - block): true if the robot arm is holding the block ?x
2. (block-clear ?x - block): true if the block ?x is not under any another block
3. (robot-hand-empty): true if the robot arm is not holding any block
4. (block-on-table ?x - block): true if the block ?x is placed on the table
5. (on ?x - block ?y - block): true if block ?x is on top of block ?y

Instruction: rearrange the blocks such that block_1 is on top of block_2, block_2 is on top of
↪  block_3, and block_5 is not on the table.

Goal in PDDL:
```
(:goal
    (and
        (on block_1 block_2)
        (on block_2 block_3)
        (not (block-on-table block_5))
    )
)
```

Here is the task.

Domain information: xxx.
```