# OpenReview forum: "Leveraging Pre-trained Large Language Models to Construct and Utilize World Models for Model-based Task Planning"
_NeurIPS.cc/2023/Conference — NeurIPS 2023 poster_

### Official Review · Reviewer_iFiN · 2023-07-06

**Soundness:** 2 fair
**Presentation:** 2 fair
**Contribution:** 2 fair
**Rating:** 5
**Confidence:** 4

**Summary:**

This paper proposed to apply GPT LLMs to perform PDDL model construction from natural language description and correction with feedbacks, and then use classic planners or LLM planners to construct plans from the PDDL models. Experiments were conducted in the synthetic domains to verify the proposal and found that GPT-4 can construct PPDL models with less errors and classic PDDL planner outperforms LLM planners with the constructed PPDL models.

**Strengths:**

1. This is a new exploration of using LLMs to construct PDDL models (or symbolic world models for AI planning)
2. Experiments are conducted to compare the performance of LLM planner and classic PDDL planner on the constructed PDDL models, and find that classic planner works better on PDDL model extracted by LLMs.

**Weaknesses:**

1. No theoretical framework is proposed to link the underlying probability models of token sequences of  LLMs to the domain and action models described by formal action description languages and its formal handling (e.g. planning, validation, correction and son). Without such a theoretical formal characterization, it is difficult to evaluate the validity and applicability/boundary of this kind of approach built on top of the LLMs, prompt-engineering, chains of feedback correction and planning and so on.
2. A few experiments are given but without a systematic handling of metrics and the underlying hypotheses to be verified, and explanation to guide the application of the proposal and future research. The experimental methodology is ad-hoc.

**Questions:**

1. Please explain the underlying models of LLMs on generating the PDDL domains and the feedback-correction mechanism; at least, explain or discuss your intuition.
2. Have you deployed the plans to real-world fetch manipulator? If you did, how does it work?
3. As your scenario is with fetch manipulator, have you try to use the LLMs to generate continuous parameters for the action operators? If not, please discuss the missing steps and future work is needed.
4. What will be the solution to connect PDDL plans to real world execution? PDDL stream might be something you want to look at; explore how LLMs approach can be further extended to work with the full stack of task-and-motion planning systems like PDDL stream (https://github.com/caelan/pddlstream).
5. Please discuss the difference between GPT-4 and GPT-3.5 rendering different results in Table 1. This will be helpful guide  readers in choosing the right LLMs to apply your proposal.

---

> ### Author Rebuttal · Authors · 2023-08-10
>
> > 1. Clarification on the problem setting (Have you deployed the plans to real-world fetch manipulator? If you did, how does it work? What will be the solution to connect PDDL plans to real world execution?)
>
> We are looking at the **model-based task planning** problem, where the main goal is to get the correct domain model and get the correct task plan. Within that setting, we are using LLMs to improve the efficacy of model acquisition part. We are evaluating (1) the correctness of the initial generated model itself (2) the effectiveness of the iterative human-in-the-loop improvement of the model and (3) once the model is deemed adequate by the user, the effectiveness of using the model in generating plans (with the help of an external sound model-based task planner).
>
> The goal of obtaining correct task plans is in line with the objectives of the current literature (that leverages LLMs for task planning [1,2,3,4]). There are several standard frameworks to connect correct task plans to execution on specific robotic/embodied agents [5,6], but this is not in the scope of this work and the targeted literature.
>
>
> [1] Yao, Shunyu, et al. "ReAct: Synergizing Reasoning and Acting in Language Models." ICLR 2022.
>
> [2] Huang, Wenlong, et al. "Language Models as Zero-Shot Planners: Extracting Actionable Knowledge for Embodied Agents" ICML 2022.
>
> [3] Valmeekam, Karthik et al. “On the Planning Abilities of Large Language Models (A Critical Investigation with a Proposed Benchmark).” arXiv:2302.06706
>
> [4]  Liu, Bo, et al. "Llm+ p: Empowering large language models with optimal planning proficiency." arXiv:2304.11477
>
> [5] Ghallab, Malik, Dana Nau, and Paolo Traverso. Automated Planning: theory and practice. Elsevier, 2004.
>
> [6] Garrett, Caelan Reed, et al. "Integrated task and motion planning." Annual review of control, robotics, and autonomous systems 4 (2021).
>
> > 2. No theoretical framework is proposed to link the underlying probability models of token sequences of LLMs to the domain and action models described by formal action description languages and its formal handling (e.g. planning, validation, correction and so on).
>
> We believe that we are very much in tune with the current practice of carefully doing empirical evaluations on the outcomes.  While it would indeed be great to have theoretical models from the token probability level to explain what LLMs are capable of doing, we note that almost none of the work about the LLM behavior–from the AGI sparks [1] to the various planning capabilities papers [2,3] –are able to offer such models at this point.  We do note that our outcome evaluation does leverage formal checks, such as the syntax verification checks by VAL, the PDDL validator.
>
> [1] Bubeck, Sébastien, et al. "Sparks of artificial general intelligence: Early experiments with gpt-4." arXiv:2303.12712.
>
> [2] Valmeekam, Karthik et al. “On the Planning Abilities of Large Language Models (A Critical Investigation with a Proposed Benchmark).” arXiv:2302.06706
>
> [3] Huang, Wenlong, et al. "Language Models as Zero-Shot Planners: Extracting Actionable Knowledge for Embodied Agents" ICML 2022.
>
>
> > 3. Please explain the underlying models of LLMs on generating the PDDL domains and the feedback-correction mechanism; at least, explain or discuss your intuition.
>
> In studies spanning various research areas, LLMs have demonstrated strong capability in self-correcting their output by leveraging critic feedback. For instance, in program synthesis, LLMs can rectify programs by iteratively incorporating error messages generated by compilers or interpreters. In our PDDL generation task, we rely on the same type of self-correction ability of LLMs. For detailed descriptions and examples of feedback templates, we recommend the reviewer to check out our Appendix, especially Sections A.3, A.7, A.9, and A11.
>
> > 4. Please discuss the difference between GPT-4 and GPT-3.5 rendering different results in Table 1. This will be helpful guide readers in choosing the right LLMs to apply your proposal.
>
> We compared the results of GPT-3.5 and GPT-4 in the experiment section. In our conclusion, specifically around lines 296-298, we stated that GPT-3.5 demonstrates a wide range of errors with no discernible patterns regarding error types and their frequency of occurrence. This suggests that GPT-3.5 is barely up to the task of PDDL generation, and hence GPT-4 stands as the only publicly accessible LLM that possesses sufficient capabilities at the time of submission.

---

> > ### Comment · Reviewer_iFiN · 2023-08-20
> >
> > Thank you for responding to my review. I've updated my score slightly. At this point, I'd like to encourage the community to not to stop at the practice of empirical evaluations but dive deeper into more formal methods or more formally connecting to formal methods, e.g. the soundness, completeness and other formal properties as the classical planning community did before. Having LLMs creating PDDL kind of ADLs is worth exploring the way back to the formal capability of the symbolic logic based ADLs.

---

### Official Review · Reviewer_bNZu · 2023-07-06

**Soundness:** 3 good
**Presentation:** 3 good
**Contribution:** 3 good
**Rating:** 6
**Confidence:** 4

**Summary:**

This paper proposes a translation approach to using LLMs for planning. Instead of relying on LLM prompt completion to generate plans, this approach parses a written description of the domain into a valid PDDL domain description. The user can then opt to use the PDDL domain description with an automated PDDL planner, or use the domain description along with prompting to force an LLM to generate a plan that can be parsed by a plan validator.

The evaluations were based upon International planning competition domains that are produced to push the limits of symbolic planning domains and are generally considered as challenging benchmarks within the symbolic planning community.

The key findings of the paper can be summarized as follows:
1) Allowing LLMs to generate PDDL descriptions working in conjunction with syntax validators and human experts to create a valid PDDL description. The process was still too error-prone to completely automate without human supervision.
2) LLMs are not capable of planning even with the restricted PDDL grammar for symbolic planning domains representative of the state-of-the-art in symbolic planning community.
3) External symbolic modules that validate plans are capable of improving LLM planning performance, but is still well below the performance of a purpose-built automated planning algorithm

**Strengths:**

The paper appropriately evaluates the capability of LLMs to plan and reason with domain sizes representative of the state-of-the-art in symbolic planning. As can be seen from table 1, the domains are significantly larger than ones seen in evaluating LLMs for planning. both in terms of number of actions and in terms of number of literals within the domain.

The claims of the paper are appropriately scoped, and are well supported by the experimentation. Care was taken by the authors to explicitly characterize the difficulty of the planning domain.

The key results of the paper are intuitive, in that it appears that LLMs are capable of translating information from one format (textual descriptions) to another (formal planning domain predicate and action description), but perform poorly at the actual task of planning and reasoning. Appending plan validator feedback to the prompt and replanning with an LLM is akin to backtracking in planning algorithms but is a very roundabout and computationally inefficient way of implementing search algorithms. It is essentially using the LLM output and context window as memory.

**Weaknesses:**

Reliance on the user: The LLM is forced to generate the domain description one action at a time. This pushes the workload of identifying the right number of actions and stopping criterion onto the user. understanding whether the domain description if complete and consistent is a major aspect of generating the PDDL domain, and right now it seems to be done by the user.

Explicitly stating the contribution: I would like the authors to reflect and restate what the major contribution of this paper is. Is it a prompt structure that helps LLMs consistently translate descriptions into PDDL. Or is it the definitions of the symbolic modules used to validate the planning module developed in particular to work well with LLMs. The statement of why the paper is novel, and significant seems missing from the submission, though I appreciate the value that this work brings to the table.

LLMs are new on the scene, and I have seen quite a few papers in the LLM + symbolic planning that are very recent and potentially close to the NeurIPS submission deadline. I would hope that if accepted, the authors acknowledge this work and rewrite the background and related work sections accordingly [1],[2]

[1] - Liu, B., Jiang, Y., Zhang, X., Liu, Q., Zhang, S., Biswas, J. and Stone, P., 2023. Llm+ p: Empowering large language models with optimal planning proficiency. arXiv preprint arXiv:2304.11477.
[2] - Valmeekam, K., Olmo, A., Sreedharan, S. and Kambhampati, S., 2022. Large Language Models Still Can't Plan (A Benchmark for LLMs on Planning and Reasoning about Change). arXiv preprint arXiv:2206.10498.

**Questions:**

Did the authors evaluate whether the LLM can generate PDDL domains of the same description at various levels of action abstraction? If so I am missing that from the draft.

What is the role of the user in the proposed system?

How does your proposed framework decide if the planning domain generated is adequate?

---

> ### Author Rebuttal · Authors · 2023-08-10
>
> > 1. Some papers in the LLM + symbolic planning
>
> In fact, we cite both papers and discuss them in detail in our submission:
>
> (a) LLM+P: starting from line 346, we explained the fact that  LLM+P only uses LLMs for translating user instruction into PDDL format. In contrast, our work does much more than such translation (e.g., deriving predicates and constructing domain models)
>
> (b) "Large Language Models Still Can't Plan": we have cited a newer version of this "can't plan" paper [1], authored by the same group of researchers. Their study serves as a significant motivation for our work.
>
> [1] Valmeekam, Karthik, et al. "On the planning abilities of large language models (a critical investigation with a proposed benchmark)."
>
> > 2. Is the LLM merely doing the translation? Various levels of action abstraction?
>
> We would like to emphasize that our framework uses LLMs beyond simple translation. For comparison, we list here two well-known papers that merely do the translation [1,2]. Both of them **require a PDDL domain model to be given**, and then they use LLMs to translate natural-language user instruction and detailed state description to PDDL goal specification and init states.
>
> In contrast, our framework requires the LLM to derive the set of predicates and to determine the preconditions and effects by taking partial information from the action description and leveraging the web-scale knowledge within the LLM. It is crucial to note that we do not assume the action descriptions are provided as a comprehensive list of preconditions and effects that can be directly mapped to their PDDL counterparts. The LLM needs to supply the input information with its own knowledge. In this regard, we believe our task is more akin to a program/code synthesis task rather than a mere translation task.
>
> Also, as explained in the paper starting from line 299, we did conduct extra experiments with action descriptions at a different level of abstraction. Results confirm that GPT-4 can provide meaningful knowledge acquisition when minimal information is provided in action descriptions. The reviewer may find the descriptions we used in the Appendix starting from Section A.6.3, wherein the minimal action descriptions are given in the “Action description” box and additional details are given in the “Additional information from the user.”
>
> [1] Xie, Yaqi, et al. "Translating natural language to planning goals with large-language models." arXiv:2302.05128
>
> [2] Liu, Bo, et al. "Llm+ p: Empowering large language models with optimal planning proficiency." arXiv:2304.11477
>
> > 3. Reliance on the user
>
> As in any use case of LLMs, LLMs are only **approximately omniscient**, and without external intervention, we can’t get guaranteed correct outputs. Our whole point is that it is easier to have the human intervention be limited to getting the model correct, and to let the sound planner ensure plan correctness, than it is to have humans in the loop for checking every single plan’s correctness.
>
> We understand that any system would benefit from a higher degree of automation, but we believe the existing literature on leveraging LLMs for task planning tends to overlook a critical problem of how the generated (potentially erroneous) plans are going to be used --- most existing papers only end up reporting a subpar success rate/plan correctness rate. Though not explicitly mentioned in relevant papers, the users have to eventually review and correct the plans before executing them.
>
> In contrast, what we present is a **more complete & closed-loop framework** that answers how we can obtain the final corrected plans. In fact, we posit that our approach even requires less human involvement compared to existing methods. This is because our system only necessitates the correction of the domain model once, whereas with other approaches, the user needs to inspect and correct **every plan** generated by LLMs.
>
> > 4. Explicitly stating the contribution
>
> Our contributions are as follows:
>
> (a). Our framework introduces a systematic approach to prompt LLMs for constructing PDDL domain models. This includes the entire workflow/algorithm, the prompt templates, and the management of derived predicates.
>
> (b). To ensure the syntax and factual correctness of the domain model, we utilize multiple sources of feedback, including the PDDL validator in VAL and human feedback. Additionally, we provide prompt templates to incorporate feedback into the generated models.
>
> (c). We explore two ways of utilizing the generated models: planning with an external planner, and back-prompting LLM planners with information on unmet preconditions and goals.
>
> > 5. What is the role of the user in the proposed system? Are they responsible for identifying the right number of actions and the stopping criterion?
>
> Please note that in our framework, the user is not responsible for determining the right number of actions. As detailed in the problem setting, this paper operates under the assumption of a fixed-size library of actions (which is a common setting in the existing literature).
> Overall, the user is responsible for (a) making sure the domain model is producing desired policy/behaviors, and identifying factual errors if needed; (b) providing feedback on the factual errors (basically pointing out incorrect or missing preconds/effects).
>
> Most crucially, once the model is deemed correct, it is handled by a sound planner, and the user is no longer in the loop.
>
> > 6. How does your proposed framework decide if the planning domain generated is adequate?
>
> In our framework, although tools like VAL help in pointing syntactical errors, it is eventually the user in the loop that deems the generated domain adequate by telling whether the generated behavior aligns with the user’s expectation or is executable or not. This is similar to how domains are made in the planning domain development literature (such as GIPO [1]).
>
> [1] Simpson, Ron et al. "Planning domain definition using GIPO."

---

> > ### Comment · Reviewer_bNZu · 2023-08-14
> > **Happy with the discussion**
> >
> > I am satisfied with the discussion in the author's response. Apologies for missing the relevant citations. I am still unconvinced on the utility of LLM-based planners if the domain definition generated is adequate for a symbolic planner, however that is not the sole contribution of this paper. I trust that the authors will revise the introduction to include the succinct version of the stated contributions, and update the presentation of method in the paper, to improve the clarity of the presentation inline with the response here. All the edits required are merely in the text, and no additional experiments are required. I am happy to update the score accordingly.

---

> > > ### Author Response · Authors · 2023-08-15
> > >
> > > Thank you for your consideration and valuable comments! We'll ensure to pay attention to the suggestions in our revised version.

---

### Official Review · Reviewer_33uW · 2023-07-07

**Soundness:** 3 good
**Presentation:** 3 good
**Contribution:** 3 good
**Rating:** 7
**Confidence:** 3

**Summary:**

There's been a fair amount of recent work on using LLMs to directly write plans for planning problems. In contrast, this work divides the planning process into two stages where first an LLM is used to translate natural language descriptions of actions to formal PDDL actions, then goals can be planned towards using a classic PDDL planner like FastDownward. Alternatively, the formal model can be used to validate and propose feedback on plans proposed by an LLM planner. They also explore using feedback from an automated PDDL validator (VAL) to correct things like syntax errors, as well as human feedback, both of which are ingested by the LLM which is prompted to repair the model.

**Strengths:**

- The two stage process proposed here (first translating to a PDDL representation, then planning) is a nice setup. Unsurprisingly, when the PDDL model is correctly specified (e.g. after rounds of human feedback), the performance is much better than a setup that lacks this PDDL model construction step.
- The paper flows very nicely, giving a good overview of the problem and concise summary of the method in the intro, and a nice related work and PDDL/planning background section.
- The inclusion of detailed descriptions of the prompts is essential and much appreciated (and the full prompts given in the appendix) for fully understanding what the LLM is seeing. The authors do a great job of this. This kind of extensive appendix is invaluable for both replicating LLM work and understanding precisely what the inputs/outputs of the LLM are and its various failure modes.
- I found a lot of the analyses / results commentary throughout the paper to be quite interesting and informative, such as the comparison between GPT-3.5-Turbo and GPT-4, or the aside about how LLMs do more "reasonable" things in their plans with respect to a human-like prior, for example leaving a knife on the counter instead of placing it on top of a toaster as the symbolic planner sometimes did. These little asides gave me a better feel for the strengths/weaknesses of LLMs in the context of planning, world model construction, etc.
- The idea of translating the formal model *back* into natural language when presenting the user with errors is also a nice touch.

**Weaknesses:**

- The human feedback seems necessary given the errors GPT-4 introduces, and presumably this method wouldn't work well at all without that step of human correction. That means the method can't be fully automated (i.e. with just VAL feedback and no human feedback), or if this is the case then it isn't discussed or evaluated in the paper. Still, this is not a huge weakness in my view – methods involving humans are good too, and perhaps this could be a first work on which future work could be built that is more automated than just humans looking through a whole PDDL model trying to spot bugs. The authors discuss this a bit in section 4.2 and appendix A.4 which are quite nicely written.
- Seeing an evaluation on more domains would have been nice, but I'm not super familiar with the norms in the planning literature. What they have is enough for me to get a reasonable feel for things, so this is not a big weakness.

**Overall**, this seems like a good paper that I would weakly support acceptance on. Given the heavy involvement of humans in correcting the model I don't see this as an extremely groundbreaking result, but I think it's still a moderate to high impact well written piece of research that deserves to be published and would be appreciated by a NeurIPS audience. My opinion is definitely open to revision, and I may be missing some points as discussed in Questions below.

**Questions:**

- "GPT-3.5-Turbo produces highly noisy outputs with over 350 errors." (Line 296) I'm curious what sorts of errors are made by the model. You mention that GPT4 makes very few syntax errors (mostly reasoning errors). Is the same true of GPT-3.5-Turbo, or does it not even have syntax figured out?
- I'm curious to hear more about the *newly defined predicates* (line 187) that the LLM ended up introducing in these domains. How many new predicates does it end up producing? Is it generally obvious what the "right" predicates are, such that we'd expect it to usually match the "ground truth PDDL" (line 285) exactly, or are there situations where there could be multiple reasonable choices to make for what predicates are introduced?
- At the end of the human feedback process, are the models literally equivalent to the "ground truth PDDL" (line 285)? Or is there room for the LLM to make modeling choices that lead to different models (eg in the "newly defined predicates" mentioned above). This would be helpful for me understanding the 95% accuracy result – is it just that we've recovered the ground truth at this point by human corrections, and the ground truth can get 95%? To be clear, this is still a good result, as the idea of translating into a symbolic language for fast precise symbolic reasoning makes a lot of sense, I just want to see if I'm missing anything.
- The exclamation point on line 345 feels a little out of place with the tone, but that's just a stylistic choice
- To point out a piece of work that is quite different but seems in some ways to share in the spirit of this work – in "From Word Models to World Models" (Wong et al 2023), they similarly treat the LLM essentially as a semantic parser that translates natural language into a formal world model language (in their case, a probabilistic program) where classic tools can do most of the work. Of course, this also relates more widely to the "LLMs with tools" paradigm where LLMs call out to other libraries/simulators as in "ToolFormer" (Schick et al, 2023)  or "Mind's Eye" (Liu et al, 2022). I don't personally need to see all of this in the related work, but just a suggestion if you think it might be worth drawing the connections.
- You've got a missing bullet point in Fig 1 top left box
- I saw in Appendix A.4 you were discussing human-usable tools for debugging PDDL models – I'm curious if you've also thought about ways that this could be fully automated? For example, starting to plan with the faulty model, hitting a failure to plan, and then somehow prompting the model with this error such that it could revise the error in the model? Would love to hear if you've thought about this.

**Limitations:**

Limitations are adequately discussed by the authors.

---

> ### Author Rebuttal · Authors · 2023-08-10
>
> > 1. Involvement of humans: the human feedback seems necessary given the errors GPT-4 introduces, and presumably this method wouldn't work well at all without that step of human correction.
>
> As in any use case of LLMs, LLMs are only **approximately omniscient**, and without external intervention, we can’t get guaranteed correct outputs. Our whole point is that it is easier to have the human intervention be limited to getting the model correct, and to let the sound planner ensure plan correctness, than it is to have humans in the loop for checking every single plan’s correctness.
>
> We understand that any system would benefit from a higher degree of automation, but we believe the existing literature on leveraging LLMs for task planning tends to overlook a critical problem of how the generated (potentially erroneous) plans are going to be used --- most existing papers only end up reporting a subpar success rate/plan correctness rate. Though not explicitly mentioned in relevant papers, the users have to eventually review and correct the plans before executing them.
>
> In contrast, what we present is a **more complete & closed-loop framework** that answers how we can obtain the final corrected plans. In fact, we posit that our approach even requires less human involvement compared to existing methods. This is because our system only necessitates the correction of the domain model once, whereas with other approaches, the user needs to inspect and correct **every plan** generated by LLMs.
>
> > 2. You mention that GPT4 makes very few syntax errors (mostly reasoning errors). Is the same true of GPT-3.5-Turbo, or does it not even have syntax figured out?
>
> When the number of literals increases, GPT-3.5 starts to have problems even in following PDDL syntax. We tried to characterize the failure modes, but it turns out that there are no discernible patterns regarding error types and their frequency of occurrence. Overall, GPT-4 stands as the only publicly accessible LLM that possesses sufficient capabilities at the time of submission, and users should use GPT-4 whenever possible.
>
> > 3. Regarding newly defined predicates (line 187) that the LLM ended up introducing in these domains, how many new predicates does it end up producing? Is it generally obvious what the "right" predicates are?
>
> Please note that **all predicates** were introduced by the LLM in all domains. The "dynamically updated list of predicates" (Fig. 2) is initialized to **an empty list**. We will ensure that we provide clarification on this in our revision.
>
> Regarding the correctness of predicate design, this is indeed an open-ended problem. Our evaluation criteria are based on whether the derived predicates and the action model can precisely recover the logic of an action. For example, in the action "toggle an appliance", we expect to see there are predicates capturing the atomic events that the robot is close to the appliance and the appliance will be turned on as an effect. While there can be flexibility in naming the relevant predicates and parameters, it is crucial that the action model covers the required logic and functionality.
>
> > 4. At the end of the human feedback process, are the models literally equivalent to the "ground truth PDDL" (line 285)? This would be helpful for me understanding the 95% accuracy result – is it just that we've recovered the ground truth at this point by human corrections, and the ground truth can get 95%?
>
> We would like to clarify that, our evaluation essentially consists of three steps:
>
> (a) Step 1: evaluating PDDL model generation. We reported (in Table 1) the number of errors in the generated domain models.
>
> (b) Step 2: iteratively correcting the generated models. As described around line 330,  GPT-4 eliminated all the errors by taking human feedback. At the end of this step, we got a fully corrected domain model that 100% recovers the logic and functionality of some corresponding "ground truth model".
>
> (c) Step 3: utilizing the generated model for planning. In order to plan with a PDDL model, an external planner will also require the PDDL goal specification to be provided. So there is an extra step that uses LLMs to translate natural language instruction into symbolic goal conditions. The 5% errors (in the "95%" result) all came from this translation process (explained in line 344). It's important to note that the domain model used in Step 3 has already been corrected in Step 2, and it is deemed free from errors.
>
> We will make sure we provide better clarification on this in our revision.
>
> > 5. In "From Word Models to World Models" (Wong et al 2023), they similarly treat the LLM essentially as a semantic parser that translates natural language into a formal world model language ...
>
> We sincerely appreciate the reviewer for sharing these works, as they exhibit clear connections and a shared spirit with our work. This is a fast-growing area, and some of these works just came out after the NeurIPS deadline. When we get more space in the paper (e.g., in camera ready), we will ensure to include more up-to-date related works.
>
> > 6. I saw in Appendix A.4 you were discussing human-usable tools for debugging PDDL models – I'm curious if you've also thought about ways that this could be fully automated?
>
> Domain model inspection tools can be valuable in reducing human effort. However, as discussed earlier in Question 1, human verification and feedback remain necessary, particularly in identifying factual errors. Please note a domain model with significant inaccuracies could still generate plans without triggering errors in search algorithms. However, these plans would be completely wrong in terms of factuality (correct executability) and would deviate greatly from the real world.
>
> Nevertheless, we agree that investigating how to increase the degree of automation with existing tools can be a very promising follow-up direction.

---

> > ### Comment · Reviewer_33uW · 2023-08-16
> >
> >
> > > Please note that **all predicates** were introduced by the LLM in all domains. The "dynamically updated list of predicates" (Fig. 2) is initialized to **an empty list**. We will ensure that we provide clarification on this in our revision.
> >
> > I hadn't realized this, and it makes the results even more convincing – clarifying this in the paper would certainly be worthwhile. Before now, I had assumed you gave it some primitive predicates to work with and it did the rest. So now I realized that in the prompt of figure 2 where it says "You can create and define new predicates, but you may also reuse the following predicates:" that must be a list of predicates that came from a previous iteration, and on the first iteration this would just be an empty list or not included at all in the prompt.
> >
> > > At the end of this step, we got a fully corrected domain model that 100% recovers the logic and functionality of some corresponding "ground truth model".
> >
> > Thanks for this clarification, that's helpful.
> >
> > > The 5% errors (in the "95%" result) all came from this translation process (explained in line 344).
> >
> > Ah I see this now, thank you!
> >
> > On human involvement: It certainly would be nice to not have to rely on human verification of specs (given how difficult it can be for even humans to verify formal specs – I've worked with widely-used domains that have missing preconditions that allow for confusing non-physical actions), but I realize that is more of a grand challenge in PDDL domain generation and is out of the scope of this paper. This paper makes valuable contributions that are worth publishing regardless.
> >
> > Thank you for the thorough responses and clarifications, the discussion has strengthened my view in the paper and I have raised my rating accordingly.

---

> > > ### Author Response · Authors · 2023-08-19
> > >
> > > Thank you for recognizing the value of our work and raising the score. We sincerely appreciate your valuable suggestions. In our revised version, we will focus on enhancing clarity and providing a more detailed discussion on the benefits and potential complexities of human verification for formal specs.

---

### Official Review · Reviewer_QSdH · 2023-07-10

**Soundness:** 3 good
**Presentation:** 3 good
**Contribution:** 3 good
**Rating:** 7
**Confidence:** 5

**Summary:**

The submission investigates whether LLMs are able to create PDDL domains and problems description in three domains, two standard planning benchmarks, and one more challenging domain. The paper considers two related to the title: a) direct construction of the PDDL with an LLM, b) correcting errors in PDDLs. The paper also considers extracting plans with the obtained PDDL domains. The baseline is using a classical planner, that succeeds if the PDDL is correct. In c), they compare using the LLM as a planner, or using a ReAct style idea, but obtaining the prompts from the PDDL. This middle-ground scenario might be appealing for cases where the planning domain might not capture some elements of the problem, but an LLM can have additional insights.   The experiments are performed using GPT-4.

**Strengths:**

- The benchmarks used for experiments are relevant and not trivial. The household domain illustrates might be convincing for those not familiar with the planning literature.
- Using the symbolic models as targets instead of solving the task using an LLM can help to bring the gap between LLM and symbolic models.
- Producing a PDDL is harder than the typical implementation of LLMs plug-in. For instance, an LLM producing simple SQL might be inspected immediately, while isolated actions are not so easy to check.

**Weaknesses:**

- As the submission says, producing more complex PDDLs might be challenging.
- Both the planning benchmarks and the household domain might actually be easy for an LLM than domains in practical applications. That should be discussed further in the paper.
- The results of the manual evaluation might not extrapolate to other domains, and verification by a simulator might not be available. The submission should elaborate on the issue of explainability that is largely ignored in the document. The literature on explainability for planning should be discussed, including issues like model reconciliation.


**Questions:**

- I understand that VAL was only used for checking syntax, but VAL could also be used to validate if a plan can be executed. Was that direction studied? It seems that was easy to test as providing a plan is not much harder than the information provided to build the models.
- Which part of Fig 1 corresponds to previous approaches and which part corresponds to approaches proposed in the submission?


**Limitations:**

The work offers a methodology for building PDDL models using LLM. While the study is very preliminary, the quality is not lower that largely influential recent work like ReAct. The tasks per se is more ambitious, so further investigation is needed.

The paper ignores issues of explanability and other relevant tasks like state tracking. An incorrect model that partially tracks a state might still be useful. For instance, it might help to generate hypothesis for downstream tasks.

---

> ### Author Rebuttal · Authors · 2023-08-10
>
> > 1. Both the planning benchmarks and the household domain might actually be easy for an LLM than domains in practical applications. That should be discussed further in the paper.
>
> We would like to highlight that our evaluation domains pose significant challenges for LLMs (if we employ them directly as planner like the baselines). A recent study that delves into the planning ability of LLMs [1] reveals low correctness rates for GPT-4 generated plans, even in seemingly simple and straightforward planning domains such as Logistics and BlocksWorld (correctness rates not exceeding 30% & being very sensitive to input changes).
>
> It’s hard to say whether our planning benchmarks are actually easier for an LLM than practical domains (which may have more actions and more preconditions and effects). LLMs struggle significantly when planning tasks involve extensive search and complex sub-goal interactions. It is important to note that the level of search required and the complexity of sub-goal interactions are not directly related to the number of actions or the number of preconditions & effects. Nevertheless, in general, we expect to see a significant improvement in plan correctness by employing our approach to construct domain models and employing an external planner for search.
>
> [1] Valmeekam, Karthik et al. “On the Planning Abilities of Large Language Models (A Critical Investigation with a Proposed Benchmark).” arXiv:2302.06706
>
> > 2. The submission should elaborate on the issue of explainability that is largely ignored in the document. The literature on explainability for planning should be discussed, including issues like model reconciliation.
>
> We did not include a discussion on explainability in our submission due to space limits. However, we recognize the importance of this topic and plan to address it in our revision. Specifically, we will elaborate on the following points:
> 1. As opposed to just directly querying LLMs for plans, generating behaviors with (intermediate) symbolic models offers additional opportunities for explanation, drawing upon existing works in explainable AI [1].
> 2. In cases where the user's understanding does not align with the transcribed model, debugging and model reconciliation techniques such as D3wa+ [2] can be directly applied.
>
> [1] Kambhampati, Subbarao, et al. "Symbols as a lingua franca for bridging human-ai chasm for explainable and advisable ai systems." AAAI 2022.
>
> [2] Sreedharan, Sarath, et al. "–d3wa+–a case study of xaip in a model acquisition task for dialogue planning." ICAPS 2020.
>
> > 3. I understand that VAL was only used for checking syntax, but VAL could also be used to validate if a plan can be executed. Was that direction studied? It seems that was easy to test as providing a plan is not much harder than the information provided to build the models.
>
> We would like to clarify that VAL has been used in two ways in this paper:
> 1. (Section 4.2) We use VAL to Validate the syntax of the generated PDDL model
> 2. (Section 4.3 from line 245, and Section A.12.3 in Appendix) By taking a goal specification and a generated PDDL domain model, we can validate any plan derived from an LLM planner. In practice, we implement relevant functionalities with VAL.
>
> We believe the second way is what Reviewer QSdH suggests. It is important to note that VAL can only validate plans when **provided with a PDDL domain model and the goal specification**. In other words, VAL can only be used to validate plans after obtaining the PDDL models (e.g., through our proposed approach).
>
>
> > 4. As the submission says, producing more complex PDDLs might be challenging.
>
> We are not sure why this is listed in the weakness section. It is because producing complex PDDL models is challenging that the support of LLMs to acquire models in a semi-automated way, as done in our work, is a very promising direction. It allows for significantly accelerating the traditional knowledge engineering cycle of engineers interviewing domain experts to acquire domain models.
>
> > 5. Which part of Fig 1 corresponds to previous approaches and which part corresponds to approaches proposed in the submission?
>
> Divided by a vertical line, the left part "Using LLMs Directly as Planners" represents previous approaches, while the right part represents our proposed approach.
>
> > 6. The paper ignores relevant tasks like state tracking. An incorrect model that partially tracks a state might still be useful.
>
> We would like to note that the main aim of our work is to generate domain models--which are then used by a separate external planner to do actual plan generation. As such, state tracking is not quite relevant in the context of domain model generation. In our parallel work on plan generation with the help of LLMs, we have indeed experimented with state tracking (that is enabled through a chain of thought-like prompting).

---

> > ### Comment · Reviewer_QSdH · 2023-08-20
> >
> > Thank you for your answers. It addressed my concerns. I understand the space limitation. The scope of the paper is clear.

---

### Author Response · Authors · 2023-08-10

Dear Reviewers,

Thank you for your thoughtful reviews and constructive suggestions to improve our paper! We have carefully considered each point and have responded accordingly in separate notes under each of your reviews. Should you have any further queries or require additional clarification, please feel free to reach out to us.

Thank you once again for your valuable input.

Sincerely,

The authors

---

### Decision · Program_Chairs · 2023-09-21

**Decision:**

Accept (poster)

**Comment:**

This paper explores using large language models (LLMs) to generate PDDL domain models from natural language descriptions with a flavor of program synthesis. The models are then used with a classical PDDL planner or an LLM planner to generate plans.

The reviewers generally found the idea novel and the experiments properly evaluated, but had concerns about novelty given related concurrent work, reliance on human involvement, and lack of a theoretical framework connecting LLMs to PDDL. All reviewers recommended varying levels of acceptance, citing the challenging planning benchmarks used, value of translating textual descriptions into formal representations, and thorough experiments. In line with the reviewer-author discussion, I do feel that the lack of a theoretical grounding for the framework is in itself not a reason for rejection but do encourage the authors to pursue this line of thought as part of future work.

In their rebuttal, the authors clarified their contributions, justified the level of human involvement, and compared to related works published near the deadline. Additional experiments adequately addressed reviewer questions about action abstraction and user roles, resulting in several reviewers raising their initial scores.

The idea of using LLMs to construct planning domain models is novel and valuable. The experiments, while limited to synthetic domains, properly stress test capabilities and analyze errors. The reliance on human verification seems necessary for now, but could potentially be alleviated through finetuning or an RLHF equivalent. I would recommend acceptance, with revisions to further clarify novelty and contributions.